# Structural and functional properties of a probabilistic model of neuronal connectivity in a simple locomotor network

Andrea Ferrario[1]*, Robert Merrison-Hort[1], Stephen R Soffe[2], Roman Borisyuk[1]

[1]School of Computing, Electronics and Mathematics, University of Plymouth, Plymouth, United Kingdom; [2]School of Biological Sciences, University of Bristol, Bristol, United Kingdom

**Abstract** Although, in most animals, brain connectivity varies between individuals, behaviour is often similar across a species. What fundamental structural properties are shared across individual networks that define this behaviour? We describe a probabilistic model of connectivity in the hatchling Xenopus tadpole spinal cord which, when combined with a spiking model, reliably produces rhythmic activity corresponding to swimming. The probabilistic model allows calculation of structural characteristics that reflect common network properties, independent of individual network realisations. We use the structural characteristics to study examples of neuronal dynamics, in the complete network and various sub-networks, and this allows us to explain the basis for key experimental findings, and make predictions for experiments. We also study how structural and functional features differ between detailed anatomical connectomes and those generated by our new, simpler, model (meta-model).

DOI: https://doi.org/10.7554/eLife.33281.001

*For correspondence:
andrea.ferrario@plymouth.ac.uk

**Competing interests:** The authors declare that no competing interests exist.

## Introduction

Information processing in the brain is based on communication between spiking neurons that are embedded in a network of synaptic connections. Clarifying the interplay between network connectivity and functionality is a key part of understanding how the brain generates functional behaviours (*Sporns et al., 2005*; *Marder and Calabrese, 1996*). Studying this relationship is difficult because nervous system connectivity usually varies considerably between individuals. Despite this variation each individual behaves in approximately the same way, especially in the case of simple animals. This commonality of behaviour suggests that there are some fundamental organisational principles that underlie the structure of a species' nervous system. How can we identify these fundamental properties that are shared across individuals and allow the nervous system to function correctly?

In this paper, we attempt to answer this question in the case of the hatchling *Xenopus* tadpole. Whole cell recordings and anatomical measurements of neurons, combined with computational modelling, have uncovered many important details regarding the neuronal network that controls swimming in hatchling tadpoles (*Roberts et al., 2010*; *Roberts et al., 2014*). We have previously shown how modelling of the neuronal connectivity in the tadpole spinal cord and caudal hindbrain is possible through a 'developmental' approach, whereby connections between neurons are not prescribed but appear as a result of the intersection between (simulated) growing axons with dendrites (*Borisyuk et al., 2014*). This anatomical model mimics the realistic growth of axons in the spinal cord. Following biological realism, the axon growth is guided by the concentration of chemical gradients in the spinal cord. The properties of such gradients are controlled by model parameters that

have been optimized to produce the same statistical characteristics as real measurements. Other model specifications (including soma positions and dendritic extents) are assigned from the distributions of experimental data and from general biological knowledge. The model includes several stochastic components (*Borisyuk et al., 2014*; *Roberts et al., 2014*); therefore, each model simulation generates a different pattern of connectivity ('connectome'). The connectivity can be mapped onto a functional model composed of spiking units of Hodgkin-Huxley type, with parameters chosen to match known tadpole electrophysiology (*Sautois et al., 2007*). The resulting functional model reliably produces activity patterns like those seen during real swimming (*Roberts et al., 2014*). It is important to note that the anatomical model provides a way of generating many different connectomes, such that the random variation observed between generated connectomes has the same statistical properties as measurements taken from different individual animals. Here, we set out to reveal the fundamental features of the neuronal connectivity that underlie the ability of the swim network to function robustly.

We describe a new probabilistic model of connectivity, which is generalised from a large number of connectomes generated by the anatomical model. This probabilistic model is a matrix that specifies the probability of connection between each pair of neurons. Being derived from multiple biologically realistic (anatomical) connectomes, the probabilistic model reflects the anatomical structure of the biological system. An important advantage of the probabilistic model is that it is simple enough that we can analyse the properties of the model itself, rather than individual connectome realisations. We use the probabilistic model to calculate structural properties of the tadpole network. These results are general, and therefore should reflect the fundamental organisational principles that we aim to uncover here.

Graph and network theories (*Rubinov and Sporns, 2010*) are increasingly used to study connectivity of different neuronal networks: *C. elegans* (*Varshney et al., 2011*; *Kaiser and Hilgetag, 2006*), zebrafish (*Stobb et al., 2012*), cat, rat and macaque cortical structures (*Sporns et al., 2007*; *Sporns and Zwi, 2004*; *Humphries et al., 2006*). For example, it was shown that the *C. elegans* connectome is heterogeneous and has a hub structure (*Towlson et al., 2013*): most neurons have a low number of connections but there are several highly connected 'hubs'. Hubs have been identified in many brain networks and they are likely to be formed at an early stage of development (*Varier and Kaiser, 2011*). However, not all brain circuits have hubs; for example, they have not been found in the rat reticular formation (*Humphries et al., 2006*) or zebrafish nervous system (*Stobb et al., 2012*).

Using the probabilistic model, we estimate the heterogeneity (*Hu and Wang, 2008*) and connection degree distributions (*Barabasi and Albert, 1999*; *Sporns et al., 2007*; *Varshney et al., 2011*) of the tadpole's spinal cord network. We found that the generalised tadpole network is not scale-free and that hubs do not exist; therefore in this respect the generalised tadpole network differs from the *C. elegans* connectome.

A second potential advantage of the probabilistic model is that it can be used to easily generate connectome realisations by sampling from the probability matrix, without detailed simulation of neuronal growth. This enhances its potential value as a tool for studying the functional properties of the network when combined with an appropriate physiological model. Multiple functional simulations of probabilistic connectomes demonstrated a reliable pattern of rhythmic activity, qualitatively like tadpole swimming and as seen in previous modelling (*Roberts et al., 2014*). Thus, the generalised probabilistic model shares structural and functional properties with the real biological object. However, quantitative differences showed that caution is required to avoid pitfalls when employing the probabilistic approach to study real biological activity.

Specifically, we found that the variance of the number of incoming connections (in-degree) or out-going connections (out-degree) of each neuron is higher in anatomical rather than a probabilistic connectomes. As a result of this finding, we observed that the period of the rhythm was longer in probabilistic connectomes. We can explain why the generalisation process affects the swimming period, and show how it is possible to accurately predict the period of swimming using only structural properties of the connectome. We then show how, by making suitable parameter adjustments, we can match the functionality of the probabilistic connectomes to that of the animal and anatomical connectomes. This makes it possible to use the probabilistic approach as a tool for studying real biological activity as well as structural properties of networks.

Despite the differences between anatomical and probabilistic models, we demonstrate several important advantages of using the probabilistic model in comparison to the anatomical one. For example, we could predict the position of commissural interneurons (cINs) that are active during swimming, which can be difficult to explain by the anatomical model. Specifically, our simulations show that cINs in rostral positions are less likely to fire reliably than those in caudal positions. Moreover, the probabilistic model allowed us to easily design new computational experiments that helped to clarify the following experimental findings.

By studying the connectivity of CPG neurons specifically, we show that the minimal swimming subnetwork includes neurons of two types: inhibitory commissural interneurons (cINs) and excitatory descending interneurons (dINs). Similar to experiments with the surgically isolated half semi-CNS (*Soffe, 1989*), we found that the network of interconnected dINs on one body side could still generate rhythmic activity even without commissarial inhibition. It is known from experimental measurements that some dINs have both descending and ascending axons (*Roberts et al., 2010*). Our simulations of the model without ascending dIN axons show that the ascending connections play a key role in swimming and their deletion leads to pathological activity.

To summarize, in this paper we design a simple probabilistic model (meta-model) which reflects some structural features of anatomical connectomes. We also show that it can be used to study how these features relate to real behaviour by making suitable adjustments in synaptic strengths. We consider this investigation of the tadpole spinal cord as an important example of a technique that can be widely applied to study the nervous system of other animals.

## Materials and methods

### Derivation of the probabilistic connectivity model

The probabilistic connectivity model is derived from multiple connectomes generated by our existing anatomical model: a developmentally inspired model which is biologically realistic and incorporates a large number of biological measurements (*Borisyuk et al., 2014*; *Li et al., 2007a*; *Borisyuk et al., 2011*). The anatomical model simulates axon growth guided by chemical gradients, with model parameters that are chosen by fitting the generated axons to experimental measurements. As the growing axons intersect dendrites, which are allocated along the body according to experimental measurements, synapses form and make connections between neurons.

Here, we explain some details of the anatomical model that are important for understanding the new probabilistic model (for more details about the anatomical model, see Appendix 1). The anatomical model includes $N = 1382$ neurons of the seven types known to generate the swimming response. The network is divided between neurons in the sensory pathway (RB, dlc and dla), CPG neurons (dIN, cIN and aIN), and output motor neurons (mn). Sensory pathway neurons deliver sensory stimulation to CPG neurons. CPG neurons are responsible for the generation and maintenance of the swimming activity pattern. Motor neurons (mn) deliver CPG output to muscles and generate locomotion (*Figure 1*). The model is simplified by fixing the number of cells for each neural type, with neurons of each type equally divided between the left and right body sides. Simulation of the anatomical model results in a network with approximately 83,000 synapses on average. For a full description of the anatomical model and its implementation, see (*Borisyuk et al., 2014*; *Roberts et al., 2014*).

Importantly, the anatomical model includes stochastic components, so repeatedly running the model produces different connectomes with different numbers of connections and connection distributions. In particular, rostro-caudal coordinates of neurons can vary between connectomes. However, since the number of neurons of each type is kept constant it is possible to find a one-to-one correspondence between any two generated connectomes. First, we ordered the cell types (RB, dla, dlc, aIN, cIN, dIN, mn) and second, for each cell type we ordered neurons of that type according their longitudinal position (or the rostro-caudal (RC) coordinate) in ascending order from head to tail. For example, in any connectome neuron #1 is the most rostral RB neuron on the left side of the body, while neuron #62 is the most caudal left-side RB; neurons #63–126 are the right-side RB neurons; neurons #127–146 and neurons #147–174 are the dla neurons on the left and right sides respectively, etc. This ordering of cells is universal and does not depend on a particular connectome;

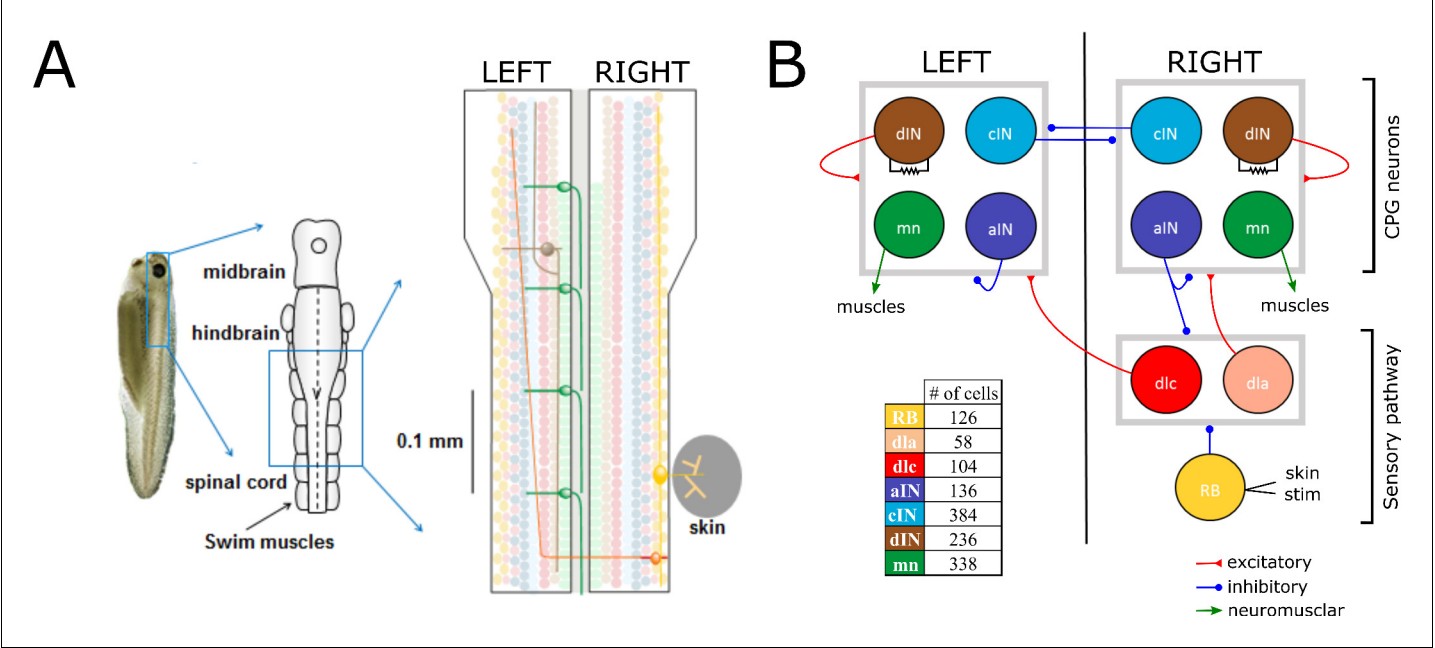

**Figure 1.** Swimming network. (**A**) Left: Photo of a 5 mm long hatchling Xenopus tadpole. Middle: two-dimensional diagram showing the indicated region of CNS seen from top with its subdivisions (midbrain, hindbrain and spinal cord). Right: Zoom of the indicated region of hindbrain and rostral spinal cord after cutting the body in half along the midline and opening it like a book. The diagram shows examples of the position of cell bodies (filled circles), dendrites (straight horizontal lines) and axons (lines extending also vertically). The floor plate separates left and right side of the CNS (grey rectangle). (**B**) Diagram showing the different populations within the swimming network and the synaptic connections between them. Connections ending on the border of each symmetrical half-centres (grey square) represent connections to any cell-type in the corresponding half-center. Descending interneurons (dINs) are locally coupled by gap junctions. Note that neuronal populations in the sensory pathway are only shown for one side of the body, but are present on both sides in the model. The table shows the colour coding and the number of neurons for each neuron type.
DOI: https://doi.org/10.7554/eLife.33281.002

therefore, we can enumerate all neurons in a universal way, providing a one-to-one correspondence between generated connectomes.

To define the probabilistic model we used the universal enumeration of neurons and considered the matrix of probabilities $P$ where $p_{i,j}$ is the probability that there is a synaptic connection from neuron $i$ to neuron $j$, $i = 1, 2, \ldots, N; j = 1, 2, \ldots, N$. Here $N = 1382$ is the total number of neurons. We defined the random Bernoulli variable $X_{ij} \in \{0, 1\}$; where $X_{ij} = 1$ means that there is a directed connection from $i$ to $j$ and the probability $\Pr\{X_{ij} = 1\} = p_{i,j}$. To calculate an estimate of this probability ($\hat{p}_{ij}$), we generated $K = 1000$ connectomes and calculated the frequency of appearance of this directed connection: $\hat{p}_{ij} = \frac{M}{K}$, where $M$ is the number of connectomes with a connection from neuron $i$ to neuron $j$. The RC-coordinate of each neuron is defined by the averaging the RC-coordinates across the $K$ generated connectomes.

The central limit theorem provides the error estimation of each entry of the probability matrix $p_{ij}$: the length of the binomial confidence interval with 95% confidence is given by $e_{ij} \approx 2 \frac{1.96}{\sqrt{M}} \sqrt{\hat{p}_{ij}(1 - \hat{p}_{ij})}$. The maximum of this error's estimate corresponds to $\hat{p}_{ij} = 0.5$, therefore, $\max_{i,j}(e_{ij}) \approx 0.06$.

The probabilities of directed connections between all neurons of the swimming network are shown in *Figure 2A*. All probabilities are between $\hat{p}_{ij} = 0$ (no connections) and $\hat{p}_{ij} = 0.69$. To visualize these probabilities we use a greyscale, where black color corresponds to $\hat{p}_{ij} = 0$ and bright pixels to high probabilities. Note: here and below we use the same notation $p_{i,j}$ for the probabilities and their estimates.

As an example, *Figure 2B* shows the sub-matrix corresponding to aIN-aIN connections. There is a black diagonal line which results from the fact that neurons cannot make connections with themselves. In fact, similar almost-diagonal lines can be seen in all of the other sub-matrices due to a

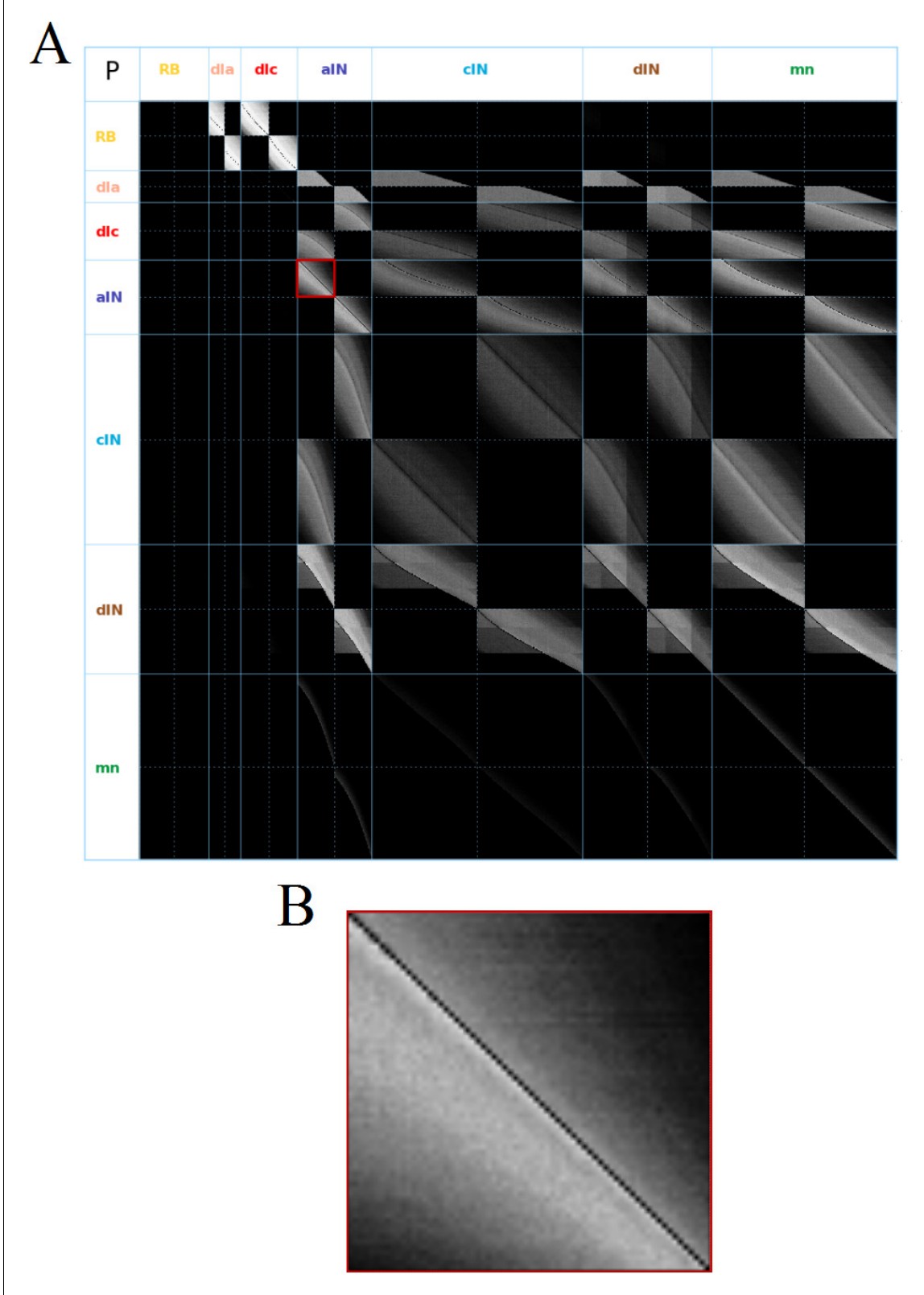

**Figure 2.** Visualization of the probability matrix P. (**A**) Image representation of the complete matrix $P$, where the greyscale intensity of the pixel in row $i$ and column $j$ represents the value of the probability $p_{ij}$. Black intensity corresponds to connection probability zero and grey intensity close to white corresponds to connection probability one. Rows and columns corresponding to neurons of each of the seven types are separated by solid blue lines. These lines separate the matrix into symmetrical sub-blocks. Within each sub-block vertical and horizontal dotted lines separate the left body side (top

*Figure 2 continued on next page*

*Figure 2 continued*

rows and left columns) from the right body side (bottom rows and right columns). In each sub-block neurons are ordered according to increasing rostro-caudal position B. Zoom of the left body side aIN→aIN sub-block (marked by a red square in A).

DOI: https://doi.org/10.7554/eLife.33281.003

feature of the growth model that prevents neurons contacting very nearby neurons. Close to the diagonal line in *Figure 2B* the shading is very bright, but this fades to black away from the diagonal. This results from the fact that the probability of two neurons being connected decreases with the distance between them. For aINs, the shading is brighter below the diagonal line, which reflects the fact that their axons are mainly in the ascending direction, making a given aIN more likely to contact aINs that are located more rostrally. While the aIN-aIN example is relatively simple to understand, neurons with more complicated growth patterns have sub-matrices with more complex structure – for example in the case of dIN-dIN connections.

The matrix *P* can be used to generate a specific adjacency matrix of directed connections (connectome) $A = (a_{ij})$ where $a_{ij} \in \{0, 1\}$ and $a_{ij} = 1$ indicates existence of the connection from neuron $i$ to neuron $j$. This matrix *A* is a particular realization of independent Bernoulli variables. We then used these specific adjacency matrices ('probabilistic connectomes') to explore their functional properties by mapping the connectomes onto our functional model to study the spiking activity in the swim network in response to stimulation.

## Functional model of spiking activity

To investigate the relationship between the network's structure and functionality it was necessary to simulate the spiking activity using the connectomes generated by the probabilistic model. We produced specific adjacency matrices from the probability matrix and used them in a functional model to simulate responses to stimulation and study spiking activity patterns. The functional model included conductance-based single-compartment neurons of Hodgkin-Huxley type with synaptic and axonal delays. In addition to the chemical synapses that are generated by the anatomical or probabilistic models, we also included the effects of electrical coupling (gap junctions) between dINs that are in close proximity to each other. We follow previous experimental (*Li et al., 2009*) and modelling (*Hull et al., 2015*) studies that have suggested that these electrical connections are an important functional property of the dIN network. Synaptic strengths, membrane channel conductances and neuron capacitances were all based on experimental results and then randomised according to a Gaussian distribution. A complete description of the functional model is given in (*Roberts et al., 2014*).

Simulations were performed using NEURON 7.3 (*Carnivale and Hines, 2006*) (RRID:SCR_005393) with a fixed time-step of 0.01 ms.

Details about the functional model and parameter values are given in the Appendix 2. The code for the anatomical, probabilistic and functional models, and the code for reproducing all the figures in this manuscript are available in Model DB from https://senselab.med.yale.edu/ModelDB/enterCode.cshtml?model=238332.

## Results

### In- and out-degrees derived from the probabilistic model

One way the structure of a network can be measured is by calculating the number of incoming and outgoing connections each element in the network has. In this section, we use the probabilistic model to calculate the mathematical expectation of the incoming connection number (in-degree) and outgoing connection number (out-degree) (*Bullmore and Sporns, 2009*) for the whole network and for different sub-networks.

Based on the assumption that the probability matrix $P = (\hat{p}_{ij})$ consists of independent Bernoulli random variables, the mathematical expectation of the in-degree $I_j$ and the out-degree $O_j$ for neuron $j$ are given by the following formulas:

$$<I_j> = \sum_{i=1}^{N} p_{ji} \qquad <O_j> = \sum_{i=1}^{N} p_{ij}. \tag{1}$$

These formulas follow from the fact that the random variables $I_j$ and $O_j$ have the Poisson binomial distribution (*Sprott, 1958*). Similarly, the formulas for the standard deviation of these random variables are the following:

$$std(I_j) = \sqrt{\sum_{i=1}^{N} p_{ji}(1-p_{ji})} \qquad std(O_j) = \sqrt{\sum_{i=1}^{N} p_{ij}(1-p_{ij})}. \tag{2}$$

*Figure 3A* shows the mathematical expectation and standard deviation (calculated using formula (1) and (2)) of the in-degree (upper panel) and out-degree (lower panel) for each neuron on each body side according the RC-coordinate (the equivalent figure for the right side is very similar and omitted here). For each cell type, the shape of the in- and out-degree distribution is very specific and depends on the soma position. For example, motor neurons (green), have high in-degree and very low out-degree. Almost all shapes are unimodal with a skewed position of the maximum. This is

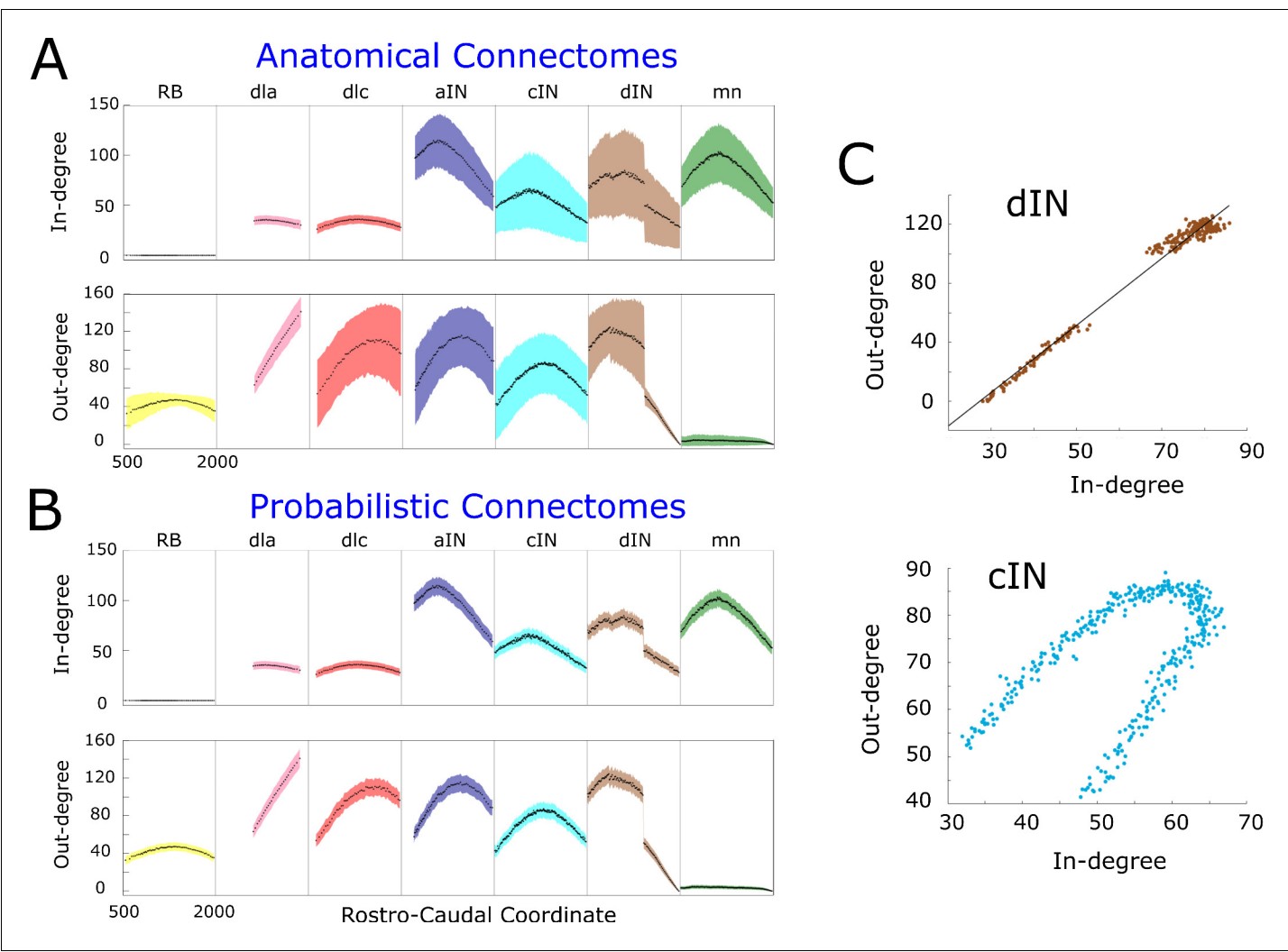

**Figure 3.** In- and out-degrees. (A-B) Average in/out-degree and standard deviation for each cell in anatomical (A) and probabilistic (B) connectomes. Neurons are divided by cell type and their degrees are plotted as a function of their rostro-caudal (RC) position. (C) Scatter plots of in- vs out-degree for CPG neuron cINs and dINs (top) and cINs (bottom): light-blue and brown dots correspond to cIN and dIN neurons, respectively. Black line shows the linear regression model for dINs ($r = 0.99$).

DOI: https://doi.org/10.7554/eLife.33281.004

a consequence of the interplay between primary and secondary axons in the developmental model and the RC-coordinate distributions of their somata. The absence of descending axons for dla neurons (pink) and their 'parallel' pattern of growth leads to almost linear increase of their out-degree. A similar explanation applies to the linearly decreasing shape of in- and out-degrees for dINs near the tail, with RC-coordinates more than 1400 $\mu m$, which have only descending axons. Interestingly, aINs have high in- and out-degrees, suggesting that, on grounds of connectivity, they could play a significant role in the network activity; however, both experiments and simulations of the functional model revealed that aINs are rarely active during swimming. This emphasises the key importance of considering both structural and functional properties in network activity.

*Figure 3B* shows estimates of the mean and the standard deviation for the connectomes which are generated by the anatomical model; we numerate them by index $\alpha$ ($\alpha = 1, 2, \ldots, 1000$). For each neuron $i$ of generated connectome $\alpha$, we consider samples of in- and out-degrees:$I_i^\alpha$, $O_i^\alpha$ ($i = 1, 2, \ldots, N = 1382$). We use these samples to calculate the estimates of the mean and the standard deviation. Obviously, the average in- and out- degrees for anatomical connectomes (black dots in *Figure 3A*) are exactly the same as the mathematical expectations of the probabilistic model shown by black dots in *Figure 3B*. However, the estimates of standard deviation for anatomical model are significantly larger for many neurons. In case of dla and dlc cell types the standard deviation of incoming connections is similar for both anatomical and probabilistic models. The reason is that the neurons of these cell types receive connections from sensory rb neurons and the number of incoming connections to dla and dlc neurons has a very low variability.

The independence of in- and out-degrees when the whole network is considered together is characterised by a small value of the correlation coefficient $r = -0.07$. However, some sub-nets showed strong dependencies. Scatter plots (pairs $(I_j, \ O_j), \ j = 1, 2, \ldots, N,$ where $N$ is the number of pairs) for cINs and dINs (*Figure 3C*) showed the linear dependence for dINs ($r = 0.99$). Other neuron types, for example cINs, show some more complicated dependence.

A key structural property of a network is whether or not it is scale-free. A scale-free network contains some 'hub' nodes with large numbers of connections in comparison to other nodes, and is particularly robust to removal of random nodes (*Barabasi and Albert, 1999*). In line with the standard approaches used for analysing scale-free structures, we calculated the distributions of in- and out-degrees. We found that all these distributions are localized around the mean and they have no tail (for this reason, we do not show these distributions here). Therefore, all these networks are not scale-free. One way in which a network can be categorised as scale-free or not is by quantifying the heterogeneity of its nodes' in- or out-degrees. We calculated the so-called heterogeneity index $H$ (*Hu and Wang, 2008*) for in- and out-distributions to estimate the variability of in and out-degrees. We compute this index to confirm that it is less than the threshold for scale-free networks (*Hu and Wang, 2008*). The heterogeneity is given by the following formula:

$$H = \frac{\sum_{i=1}^{N} \sum_{j=1}^{N} |d_i - d_j|}{2N^2 \bar{d}},$$

Here $d_i$ is either in- or out-degree of neuron $i$, $\bar{d} = i \sum d_i$ is the average degree (either in- or out-), and $N$ is the number of neurons. Note, we calculate the heterogeneity index using the probabilistic model without considering any particular connectome.

*Figure 4A and B* show the value of $H$ for each cell-type and for in- and out-degrees respectively. A standard approach for determining whether or not a network is scale-free is to compare its heterogeneity index with that of a known scale-free random network. It is known that random scale-free networks with power $2 \leq \alpha \leq 3$ have $H \geq 0.3$ (*Hu and Wang, 2008*). In contrast, all heterogeneity indices that were calculated (*Figure 4*) were relatively low (for in-degrees: $H<0.2$, for out-degree: $H<0.3$); therefore, all degree distributions in the probabilistic model were rather homogeneous. From this we concluded that for each cell type the network is not scale-free, and therefore does not contain hub neurons. Thus, the connectivity of the whole tadpole spinal cord network appears organized in such a way that there are no hubs.

Another standard approach for detecting the heterogeneity and the presence of hubs is to analyse the decay of the tail of the degree distributions and compare the rate of decay with various standard functions: typically the Power Law, Exponential or Weibull distributions (*Clauset et al.,*

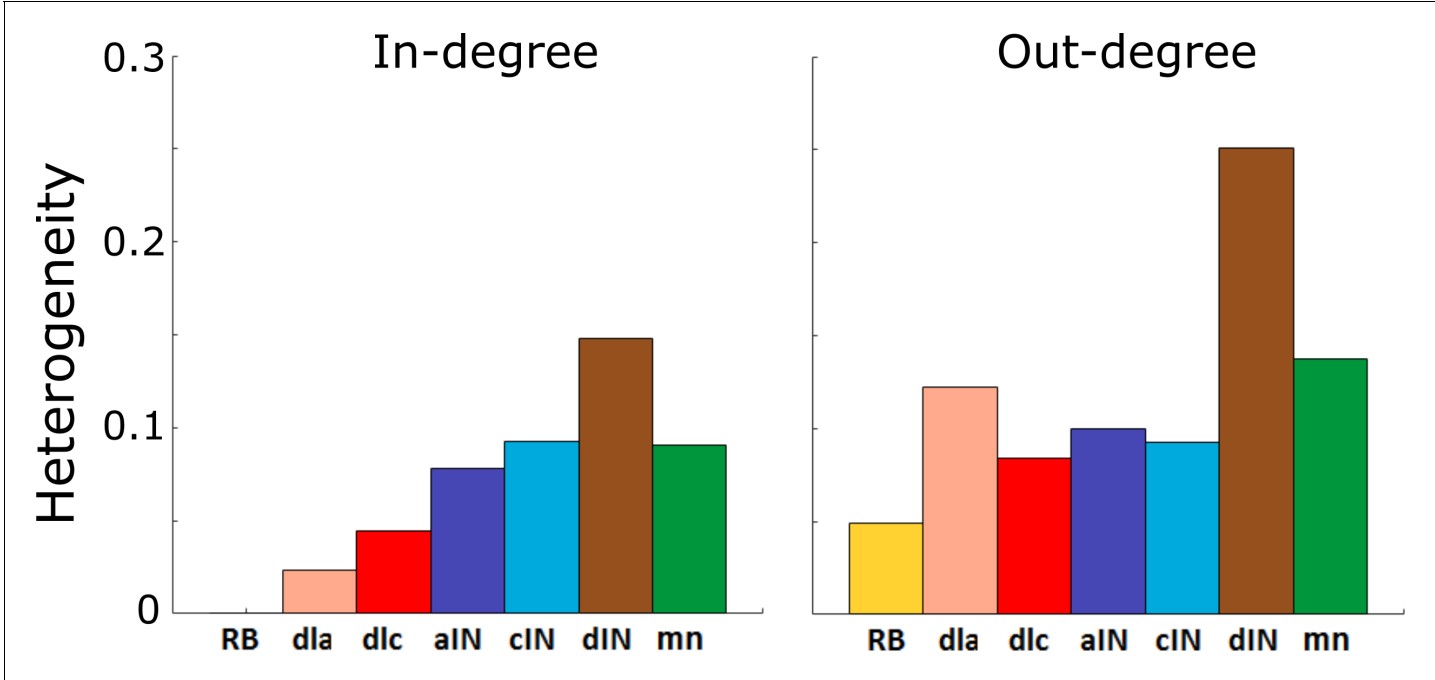

**Figure 4.** Heterogeneity index of the in-degree and out-degree distributions of each of the seven cell types.
DOI: https://doi.org/10.7554/eLife.33281.005

*2009*). As previously said, in case of the probabilistic model the in- and out- degree histograms have no tail and therefore they differ from each of these distributions. Thus, this method for heterogeneity estimation is not applicable.

## Functional properties of the model: reliable swimming

The next stage was to investigate the spiking activity of connectomes to see whether they behaved like those generated anatomically (*Borisyuk et al., 2014*) and as described behaviourally (*Roberts et al., 2014*). This was necessary to evaluate whether the probabilistic approach provided a useful tool for exploring biological function.

To investigate the spiking activity of connectomes generated by the probabilistic model, we mapped them onto a functional model composed of single compartment Hodgkin-Huxley type neurons, following the approach described in *Roberts et al. (2014)*. To simulate the basic experiment where brief stimulation of the trunk skin initiates swimming in the tadpole, we excited two adjacent sensory RB neurons on one side of the body at a randomly selected RC position. The RB activity propagates along their own axons and then in the sensory pathway (via dla and dlc neurons) to deliver excitation to CPG neurons on both sides of the body. These CPG neurons (cIN, dIN, aIN) generate a pattern of rhythmic spiking alternating between the left and right body sides suitable to drive swimming movements. We repeated this experiment 100 times using different generated adjacency matrices. We found that in all simulations the functional model produced a swimming-like pattern where: firing was rhythmic; neurons that were active fired once per cycle; firing alternated between the two sides; and firing on each cycle was most delayed towards the tail.

However, although connectomes from both the anatomical and probabilistic model produced qualitatively similar swimming activity, the probabilistic model produced a rhythm with significantly longer cycle periods ($68.6 \pm 0.8$ ms (mean $\pm$SD), range from 65 to 70 ms) than the anatomical connectomes ($58 \pm 1.8$ ms), as shown in *Figure 5A*. We investigated the underlying cause of this difference, and in doing so gained an insight into how the structure of the network affects swimming period, a key characteristic of the system's behaviour.

What determines the period of one swimming cycle? A swimming cycle starts when dINs on one side of the body spike. These excite cINs on the same side, which then spike and inhibit dINs on the

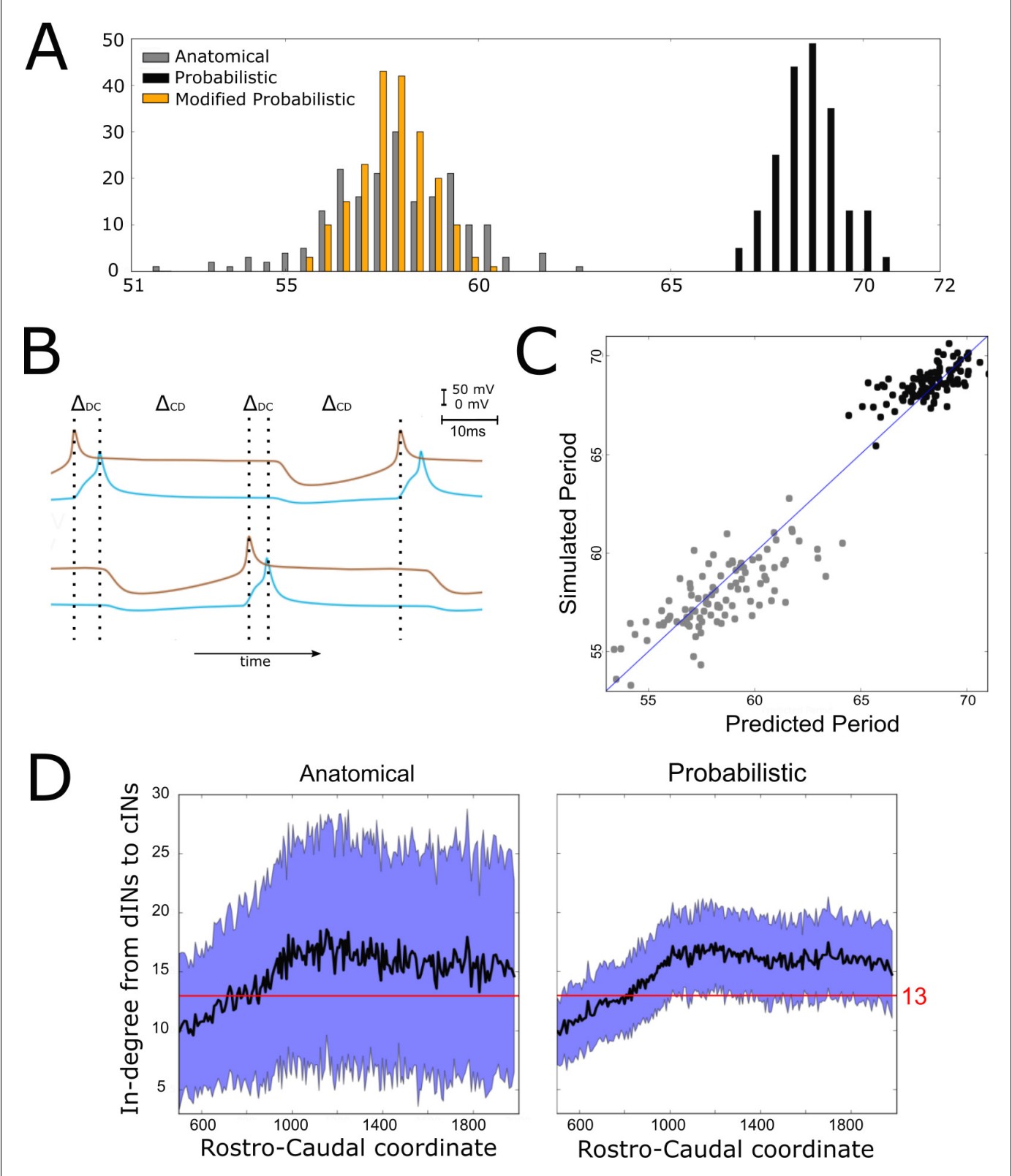

**Figure 5.** Investigating the difference in swimming cycle period between anatomical and probabilistic connectomes. (**A**) Swimming period (as defined by median motoneuron spiking period) for 200 anatomical connectomes (grey), for 200 probabilistic connectomes (black) and 200 probabilistic connectomes where cIN to dIN synaptic strength is reduced (see text for details). (**B**) Example membrane potentials of example dINs (brown) and cINs (blue) on the left and right side during one swimming cycle. The swimming period is a sum of (twice) the delay between dIN and cIN spiking ($\Delta_{CD}$ and

*Figure 5 continued on next page*

*Figure 5 continued*

(twice) the delay between cIN and contralateral dIN spiking ($\Delta_{CD}$). (**C**) Network structure allows us to predict swimming period. Each point shows for one connectome (different from those used in part C and for linear regression) the predicted period based on the connectivity, with the actual period from simulation plotted on the vertical axis. The blue line shows the case where the prediction perfectly matches the simulation. (**D**) More cINs are inactive in anatomical connectomes than in probabilistic connectomes. Although the average in-degree (black line) is similar under both conditions, the standard deviation (blue area) is much higher for anatomical connectomes. This increased variance in anatomical connectomes means that more cINs receive fewer than the 13 connections from dINs that are required for reliable spiking.

DOI: https://doi.org/10.7554/eLife.33281.006

opposite side, leading to delayed spiking of dINs on the opposite side through post-inhibitory rebound (PIR). Thus, the swimming period can be approximated as $T = 2(\Delta_{DC} + \Delta_{CD})$, where $\Delta_{DC}$ is the delay between spiking of dINs and the subsequent spiking of the ipsilateral cINs they excite, and $\Delta_{CD}$ is the delay between spiking of cINs and the subsequent PIR spiking of the contralateral dINs they inhibit (*Figure 5B*). Both $\Delta_{DC}$ and, particularly, $\Delta_{CD}$ were significantly larger with the probabilistic connectome (anatomical model: $\Delta_{DC} = 5.3ms \pm 0.4$, $\Delta_{CD} = 23.7ms \pm 0.9$, $N = 100$; probabilistic model: $\Delta_{DC} = 6.2ms \pm 0.3$, $\Delta_{CD} = 28.2ms \pm 0.4$, $N = 100$). Together these two differences account for the overall slower swimming rhythm seen with the probabilistic model, and this is largely as a result of the increased time it takes for dINs to fire PIR spikes in response to contralateral cIN input.

What, then, determines the delay between cIN spikes and contralateral dIN rebound spiking? During swimming dINs are held depolarized by summation of NMDA-receptor-mediated excitation from other dINs, and in this state inhibition from cINs can result in delayed dIN spiking as a result of PIR. Intuitively, and from past investigations, we know that this spiking delay depends on the relative strength of inhibitory and excitatory input from cINs and other dINs, respectively. We characterised the relative strength of inhibition and excitation for a given connectome by calculating the average in-degree from cINs and from other dINs. Any cINs that received fewer than 13 connections from dINs were excluded from this calculation, since, as we shall demonstrate, such cINs are likely to be inactive. We used a linear regression model where cIN-dIN and dIN-dIN in-degrees (independent variables are $I_{cIN>13}$ and $I_{dIN}$) correlate very strongly with the period of swimming:

$$T = 2.5 \cdot I_{cIN>13} - 3 \cdot I_{dIN},$$

where $T$ is the period. The coefficient of determination $R^2 = 0.96$.

We used this linear regression model to predict firing period for 200 new connectomes (100 probabilistic, 100 anatomical). The accuracy estimated using the coefficient of determination is $R^2 = 0.94$ (*Figure 5C*). We were therefore able to predict with good accuracy a key characteristic of the network's behaviour based only on its connectivity. Note that this prediction is universal, since it does not require knowledge of whether the connectome was generated using the probabilistic or anatomical model.

Why is inhibition from cINs stronger relative to excitation from dINs, and therefore swimming slower, in connectomes generated by the probabilistic model? This is a difficult question to answer completely, but much of the difference is due to the fact that anatomical connectomes have more cINs that receive fewer than 13 connections from dINs and are thus inactive during swimming (anatomical model: 168 ± 11 inactive cINs, N = 100; probabilistic model: 101 ± 8 inactive cINs, N = 100). Although the mean dIN-cIN in-degree is very similar between anatomical and probabilistic connectomes (and above the threshold of 13), the variance is much higher in the anatomical case (*Figure 5D*).

Therefore, in anatomical connectomes there are more inactive cINs. The underlying reason for this difference in variance is that in the anatomical model neurons have randomly chosen dendritic extents, sampled from the distribution of experimentally measured dendrites (see Appendix 1). This means that some neurons have small dendrites and receive very few connections, while others have large dendrites and receive very many connections. In the probabilistic case this detail is lost, as all incoming connections to a neuron are chosen completely independently of each other.

While we can explain the quantitative difference between anatomical and probabilistic models, this difference clearly illustrates that there are potential pitfalls in applying the probabilistic approach to a particular biological question, and it must be used with caution. In this specific case, there is a

problem because the reduction in dIN to cIN in-degree variance produced by the generalization process used to generate the probabilistic connectomes has asymmetric consequences. The decreased number of cINs failing to fire because of weak excitatory input (low in-degree number) is not balanced by the effect of reducing the number of cINs with very strong excitatory input (high in-degree number). This is because, above a threshold input strength, cINs only fire a single spike per cycle (see the section entitled "Reliability of cIN spiking depends on their RC-coordinate"); changing the level of excitation above the threshold value does not alter this. The result of this asymmetry is the overall increase in the number of cINs firing with the probabilistic model and hence the lengthened cycle period. To offset this consequence of the probabilistic approach, we therefore reduced the strength of cIN to dIN inhibition (from 0.435 to 0.2 nS). As predicted, this reduced the cycle period to a range overlapping the distribution produced by anatomical connectomes and matches periods seen in the real swimming behaviour (see orange histogram of swimming periods for the modified probabilistic connectomes in *Figure 5A*).

## A core dIN-cIN sub-network can generate swimming

The probabilistic approach allows us to test the reliability of network function after removal of selected connections. As an illustration, we considered a sub-network comprising only the sensory pathway (which is not active during swimming), and dIN and cIN CPG neurons. We excluded aINs and mns simply by setting the probability of connections to and from them to zero. *Figure 6* shows one simulation of the functional model containing only this sub-network. *Figure 6A,D* shows examples of voltage dynamics for individual dIN and cIN neurons on the right and left body sides, respectively; *Figure 6B,C* shows raster plots of spiking activity for all neurons on the right and left sides of the body, respectively.

The brown and light blue dots in *Figure 6B,C* show a typical pattern of anti-phase, left-right swimming activity in the dIN-cIN sub-network. We found that in 100 independent simulations (with different reduced network connectomes) swimming activity was generated that was similar to that in *Figure 6*. The swimming period in these simulations was 57 ± 0.9 ms. These values are again within the physiological range observed in experimental recordings of swimming.

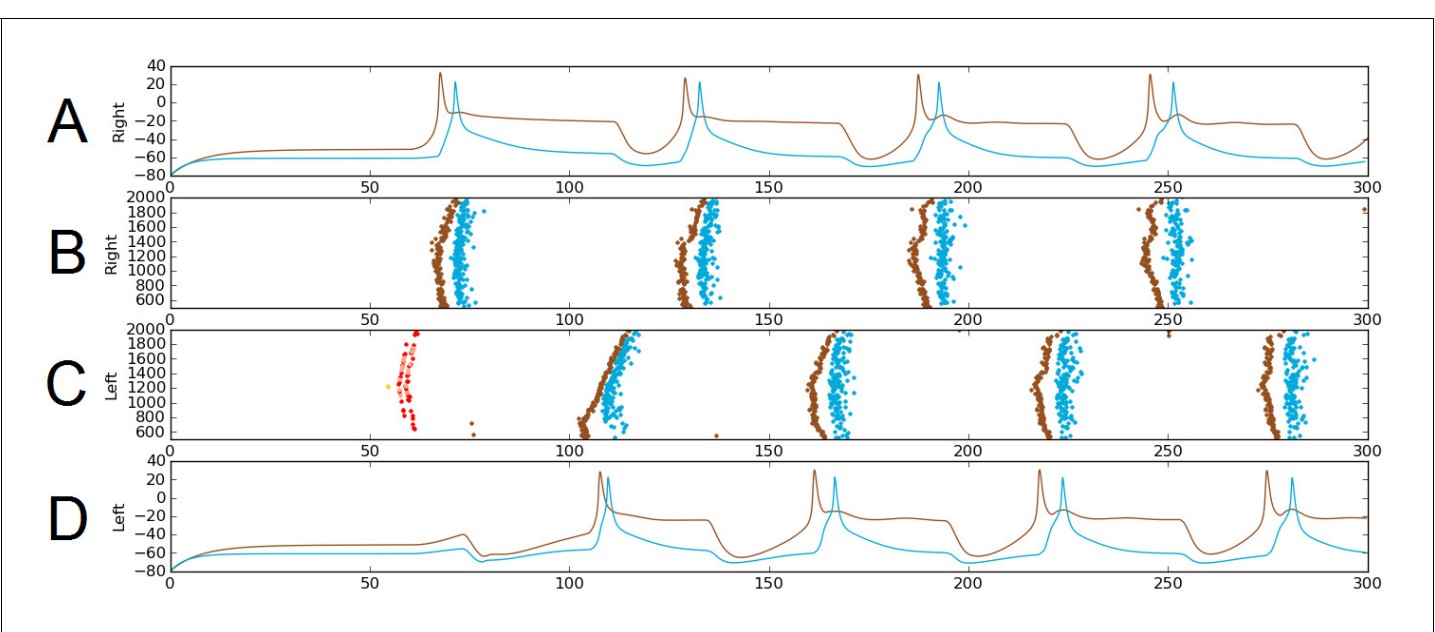

**Figure 6.** Alternating firing ('swimming') in one realization of the dIN-cIN subnetwork in a 300 ms simulation, showing activity on the right (**A–B**) and left (**C–D**) sides of the spinal cord. B and C show spike times, where the vertical position of each spike corresponds to the rostro-caudal position of the associated neuron. A and D show voltage trace examples for single selected dINs (brown) and cINs (blue) on the right (**A**) and left (**D**). Simulated sensory stimulation at 50 ms causes an RB neuron (yellow) to spike, which excites dlas and dlcs (pink and red, respectively). Excitation from these sensory pathway neurons causes the dIN and cIN neurons that form the CPG to generate an alternating rhythm.
DOI: https://doi.org/10.7554/eLife.33281.007

Previous experiments have shown that the swimming CPG includes dINs, cINs and aINs (*Roberts et al., 2010*). However, it is known that aINs have a low probability of firing during swimming, suggesting that their contributions during swimming are minimal and their role in the network is still unclear (*Li et al., 2004*). Our simulation results confirm these experimental findings by showing that the dIN-cIN subnetwork generates reliable swimming.

## Removal of commissural connections allows rhythmic firing on the stimulated body side

Experiments have revealed that an isolated side of the tadpole spinal cord without commissural connections can generate regular rhythmic spiking activity in motoneurons, with period that is lower than that of swimming (*Soffe, 1989*).

Once again, the probabilistic model readily allowed us to simulate these experimental findings by setting the probability of commissural connections from cINs and dlc neurons to zero to disconnect the two body sides. This is equivalent to a sagittal midline lesion experimentally. *Figure 7A* shows a raster plot of steady oscillatory spiking in motoneurons (green) and dINs (brown), demonstrating that the rhythmic activity was maintained and stable.

It is important to note that the mechanism that generates this single-sided rhythm is different to that which generates swimming. In swimming, inhibition from cINs causes contra-lateral dINs to fire post-inhibitory rebound spikes. In the case of separated body sides there is no cIN input to dINs, and the only other inhibitory CPG neurons, the aINs, are inactive. Instead, the rhythmic activity is caused by feedback NMDA excitation within the dIN population, as has been previously observed experimentally (*Li et al., 2010*) and in modelling (*Hull et al., 2016*). Within one simulation dINs fell into a number of different groups, based on their spiking period. In most simulations, the majority of dINs spiked rather quickly, with period approximately 24 ms (41 ± 16 dINs, N = 100 connectomes),

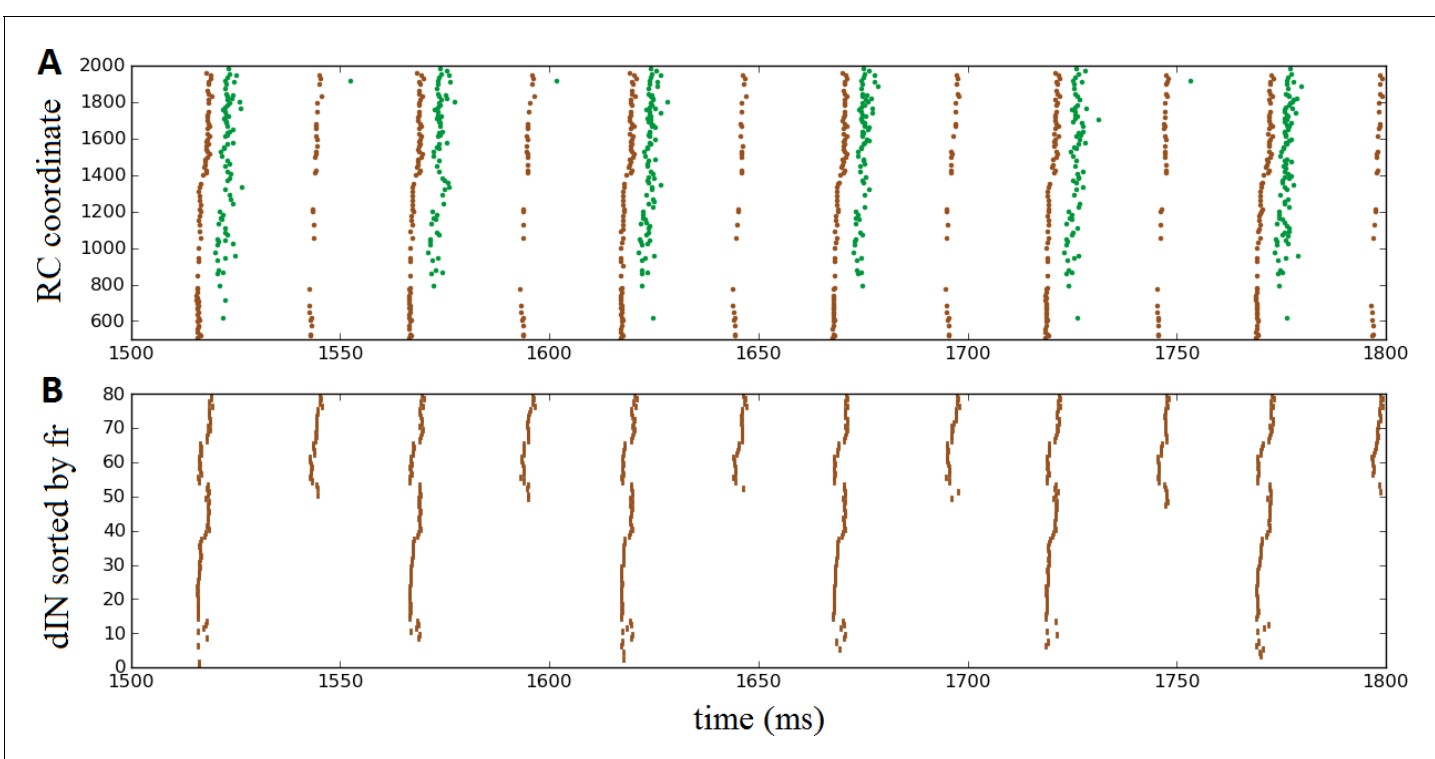

**Figure 7.** Oscillatory activity on one side of the body after removal of commissural connections. (**A**) Raster plot of spiking activity during swimming, showing dINs (brown) and motoneurons (green) on the left side of the spinal cord after removal of commissural connections. (**B**) The same dIN spiking activity as in (**A**), but with the spike trains sorted vertically based on increasing firing rate. In both cases, activity is shown between 1500 and 1800 ms post-stimulation, when the system has settled down into a stable oscillatory state.
DOI: https://doi.org/10.7554/eLife.33281.008

while most of the remaining dINs spiked with approximately double this period, approximately 53 ms (24 ± 4 dINs). A much smaller group fired twice as slowly again, with a period of approximately 101 ms (2 ± 3 dINs). *Figure 7B* makes these groups clear, by showing the same set of dIN spikes as *Figure 7A* but with the neurons sorted according to firing rate. Interestingly, motoneurons tended to fire in-phase with the intermediately sized group of dINs that spiked at approximately 53 ms (as shown in *Figure 7A*), although in some simulations some motoneurons did also spike in-phase with the faster group of dINs; further investigation is required to understand why more mns are not able to fire with the dINs in this group.

We have no direct experimental recordings of dINs following separation of the two body sides, only ventral root recordings showing motor neuron activity. In these experiments (*Soffe, 1989*), it was found that single-sided rhythmic activity was significantly faster than that seen during swimming (initial average cycle period 60 ms vs 43 ms). This was also the case with our simulations, where most mns spiked at approximately 53 ms in the single-sided cases, versus approximately 69 ms in normal swimming. From our results, we predict that recordings from dINs during single-sided rhythm generation would reveal a relatively large group of dINs that spike much more quickly than ventral root activity, and another much smaller group of dINs that fire much more slowly.

## Reliability of cIN spiking depends on their RC-coordinate

Experiments have shown that during swimming the reliability of spiking of some neuron types can vary from cell to cell (*Soffe, 1993*; *Li et al., 2007*). In simulations of connectomes generated by the anatomical model approximately 50% of cINs fire reliably, whereas in connectomes from the probabilistic model approximately 70% of cINs were reliable. Other cINs were either completely inactive or only fire on some swimming cycles. We investigated the cause of this unreliability by analysing the probabilistic model.

In the functional model, for each pair of cell types, the mean value of synaptic strength was selected in line with experimental data (*Roberts et al., 2014*) and randomised by addition of the Gaussian random variable with zero mean and relatively small variance (see Appendix 2, Synaptic Currents). In the case of synchronous bombarding, the total input to the neuron depends on both the connection strength and the number of incoming connections, therefore, the degree is an important measure. For the reliability study, we approximate the total input to cIN by the mean dIN to cIN connection strength multiplied by the mean in-degree from dINs to cINs, because dIN spike reliably and synchronously during each swimming cycle.

From simulations of 100 different connectomes, we found that the probability that a cIN spikes reliably depends on the dIN-cIN in-degree ($I_{dIN}$). If $I_{dIN} > 15$ then a cIN fires once on each swimming cycle, approximately in phase with dINs and mns on the same side; we call this a 'reliable' cIN. If $13 \leq I_{dIN} \leq 15$ then firing is irregular, meaning the cIN fires approximately in-phase with dINs and mns but on only some swimming cycles; we call this an 'unreliable' cIN. Those cINs that have $I_{dIN} < 13$ do not fire at all during swimming.

The probabilistic model allowed us to calculate the expected dIN-cIN in-degree as a function of its rostro-caudal position (*Figure 8A*). Note that this result was based only on analysis of the general probability matrix, not individual connectome realisations. The relationship allowed us to hypothesise: (1) it is likely that rostral cINs will not fire; (2) it is likely that cINs with RC-coordinate near 900 µm are unreliable, and (3) it is likely that caudal cINs will fire reliably.

To confirm these hypotheses in the model we used the results of 100 spiking simulations to calculate the probability that a cIN in a certain position will fire reliably. In *Figure 8B* we show the reliability proportion (the fraction of simulations where the cIN fires reliably) vs RC coordinate. From this, it was clear that cINs at more rostral positions have a significantly lower probability of reliable spiking than cINs in more caudal positions. Using a linear regression model, we determined a strong correlation between the cIN reliability proportion ($x$) and the average dIN-cIN in-degree ($y$) given by the linear relationship $y = 0.07 \cdot x - 0.4$ (*Figure 8C*). Note that there is currently not enough experimental data about the reliability of cIN spiking during swimming in vivo to say whether the level of cIN reliability in our simulations was realistic. However, our general results from the probabilistic model suggest that it is important that any experimental measures of cIN spiking reliability (or that of other neuron types) should take into account the rostro-caudal position of the measured cell.

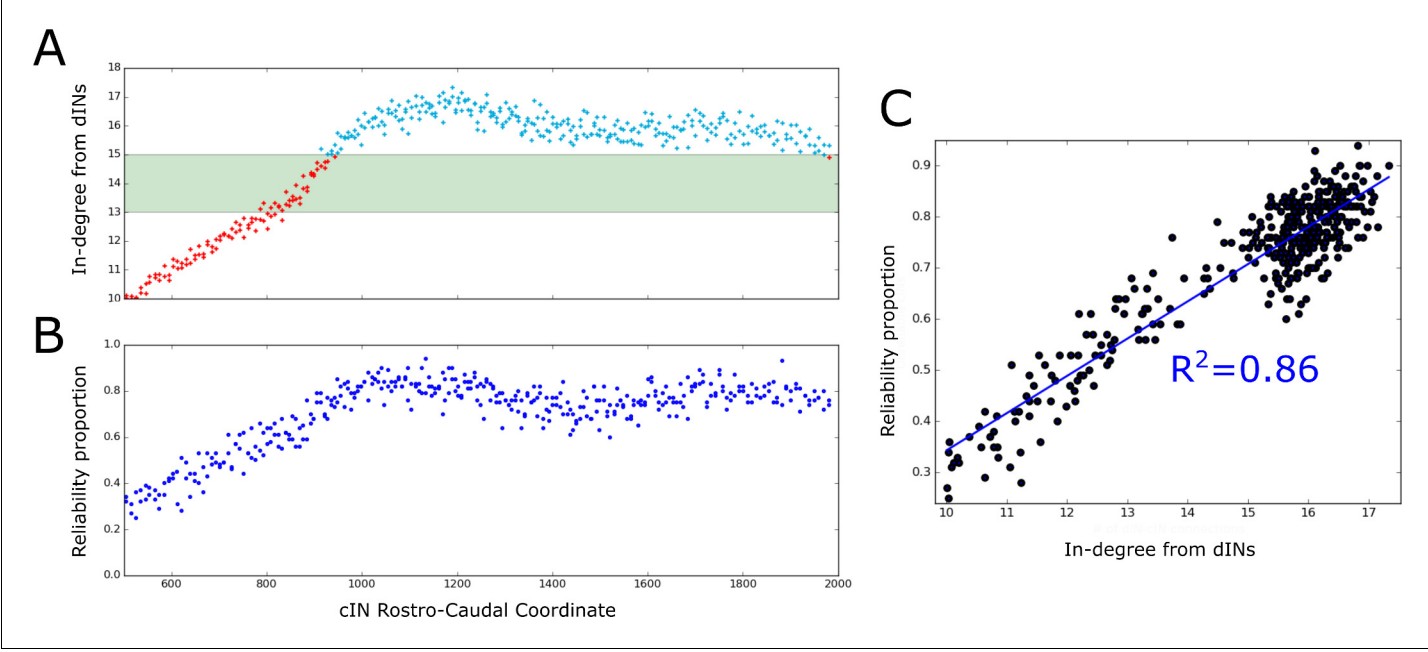

**Figure 8.** Firing reliability of cINs. (**A**) Plot of the average cIN in-degree from pre-synaptic dINs as a function of rostro-caudal position. Blue dots represent cINs that have on average 15 or more incoming connections from dINs, while red dots represent cINs that have on average fewer than 15 incoming connections from dINs. The cINs with 13–15 incoming connections (green shaded area) are most likely to fire unreliably, whereas those with fewer than 13 connections are likely to be completely inactive. (**B**) cIN reliability proportion vs cIN rostro-caudal position; for each cIN the reliability proportion is the fraction of 100 simulations where the cIN fires reliably. (**C**) Scatter plot of the cIN reliability proportion vs the average in-degree from dINs. The figure shows the linear regression line between these two variables and the corresponding $R^2$ value.

DOI: https://doi.org/10.7554/eLife.33281.009

## Ascending axons of dINs are important for swimming

It is a defining feature of dINs in the tadpole that they all have a descending axon, but some dINs which are located more rostrally have a second axon growing in the ascending direction (*Borisyuk et al., 2014*; *Roberts et al., 2014*). Simplified computational models (*Li et al., 2006*; *Wolf et al., 2009*) have shown that the swimming activity fails to self-sustain unless some excitatory interneurons have ascending connections. We used the probabilistic model to further clarify the role of ascending dIN axon branches, taking advantage of the fact that our new model allows us to run large numbers of simulations and to study the generalised connection structure. Using the probabilistic model, we removed all ascending connections from dINs and generated a modified adjacency matrix (connectome), which we then used to simulate spiking activity.

*Figure 9A* shows the in-degrees for the dIN sub-network (i.e. the number of incoming connections to each dIN from other dINs) for the standard connectome (black) and one lacking ascending dIN axons (red). In the figure, the horizontal and vertical axes show the in-degrees and the RC-coordinate of dINs, respectively. We consider here only rostral and mid-body dINs in the range of RC-coordinates from 500 to 1400 µm; more caudal dINs do not receive any synapses from ascending dIN axon collaterals, so the in-degrees are the same for both connectomes.

From *Figure 9A* it is clear that the dIN in-degrees in both cases are similar in the middle body part but are increasingly different for neurons in the rostral part. For the modified connectome, the in-degree (red dots) decays to zero in a linear way as the RC-coordinate approaches 500 µm because the dINs in the most rostral locations have a few if any connections from descending axons. As a result, the most rostral dINs in the modified connectome can only fire due to electrical coupling between dINs, resulting in the appearance of some unusual patterns of spiking activity not observed experimentally. We repeated 100 simulations of the functional model after removing ascending dIN connections. The resulting spiking activity patterns can be divided into three cases:

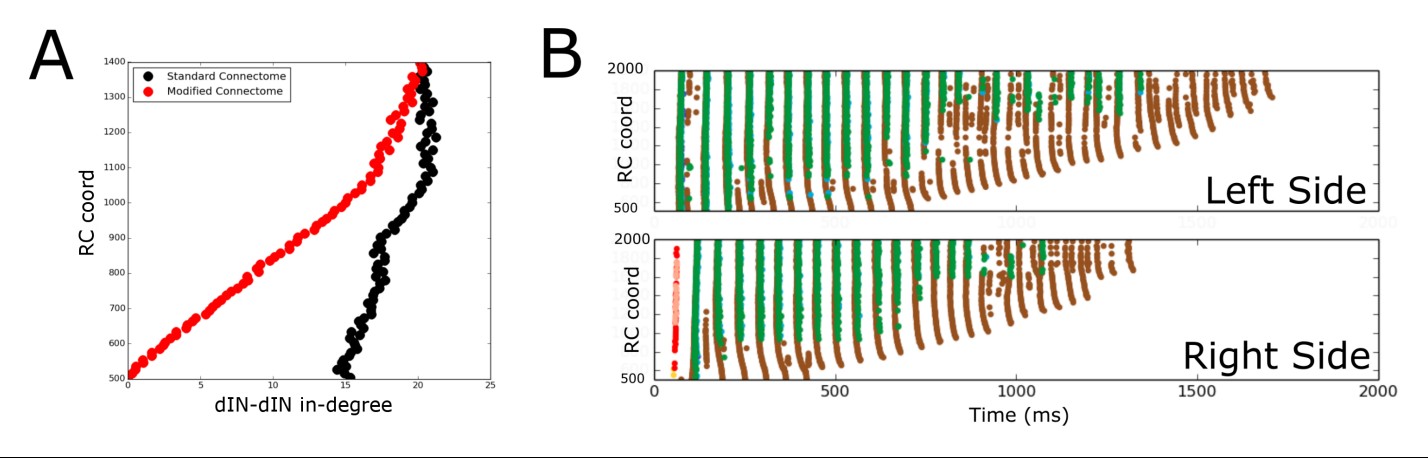

**Figure 9.** Comparison of spiking activity in the normal case and when dIN ascending axons are removed. (**A**) Average in-degree from dINs to other dINs at different rostro-caudal positions in the standard connectome (black dots) and after removal of ascending dIN axons (red dots). (**B**) Example of typical spiking activities from connectomes with ascending dIN axons removed (case 1, see text for details).
DOI: https://doi.org/10.7554/eLife.33281.010

Case 1 (63/100): In most simulations, the swimming activity was initiated but failed to persist. Swimming failures begin with rostral dINs failing to spike due to reduced excitatory drive from other dINs (ascending dIN connections are missing); this reduced excitation from the rostral dINs prevents slightly more caudal dINs from firing, and so on, as can clearly be seen in *Figure 9B*. This result is in line with previous modelling that showed that feedback excitation is a mechanism that contributes to generating persistent motor activity in a simpler model (*Li et al., 2006*).

Case 2 (36/100): In 33 of 36 simulations one side only was active. In 3 of 36 simulations both sides were rhythmically active for the total length of the simulation but they do not fire in antiphase. The pattern of spiking activity on one side is similar to the one shown in *Figure 7A*.

Case 3 (1/100): Only one simulation generated sustained swimming alternating firing between left and right sides, but the period of the oscillations was shorter than for the standard connectome (50 ms).

## Discussion

The study of neuronal connectivity is a challenging problem in contemporary neuroscience. One popular and effective method for finding cortical connectivity involved detailed tracing of a small number of individual neurons of each identified type, and then using estimates of the number of location of the different cell types to estimate complete connectivity (*Binzegger et al., 2004*). Recent development of new brain imaging techniques allows generation of 3D images of single neurons, tracing their connections and, for example, making progress towards a complete Drosophila connectome (*Lin et al., 2015*; *Shih et al., 2015*; *National Center for High-performance Computing and National Tsing Hua University, 2009*). Similar progress has been made by combining molecular, anatomical and physiological techniques to find the neuronal cell types, and connections between them, in mouse retina (*Seung and Sümbül, 2014*; *Kim et al., 2014*). Computational modelling has been successfully applied to find a sensorimotor connectome in larval Zebrafish (*Stobb et al., 2012*). In this paper, available neurobiological data have been used to describe neuronal cell types and formulate a stochastic model of connectivity, which was studied using a graph theory approach.

It is known that brain development involves multiple stochastic processes and that, in most species, individuals' connectomes are different (*Seung, 2012*). Despite differences in connectivity, most individuals under normal conditions are able to demonstrate similar functionality. This means that different connectomes include sufficient key structural features to produce a common repertoire of functionalities and behaviours. What are the key connectivity properties that define the network functionality?

Motivated by this question, we derive a probabilistic model of connectivity in the *Xenopus* tadpole CNS (caudal hindbrain and spinal cord) to study the relationship between the structure and function of the network. To derive the probabilistic model we generate 1000 connectomes using a biologically realistic anatomical model based on the 'developmental' process of axon growth (*Li et al., 2007a*; *Borisyuk et al., 2011*; *Borisyuk et al., 2014*; *Roberts et al., 2014*). A similar approach to generating connectivity from a developmental process was used by *Bauer et al. (2014)*; in this case, a reaction-diffusion model was applied to generate connectivity in a network of excitatory and inhibitory neurons with winner-takes-all functionality.

Using a universal ordering of neurons in the tadpole, we have calculated the probability of connection from each neuron ($i$) to neuron ($j$) as the frequency at which a connection exists among the thousand generated connectomes. In this way, our probabilistic model 'generalizes' structural properties of networks produced by the anatomical model.

Using the probabilistic model, we can generate an adjacency matrix representing a particular realisation of neuronal connectivity. Mapping the adjacency matrix to a functional model of spiking neurons of Hodgkin-Huxley type enables us to simulate spiking activity. We compare these simulations of the functional model to the experimental results on swimming initiated by skin touch. All generated adjacency matrixes (connectomes) mapped to the functional model generate similar swimming activity. It seems, then, that the probabilistic model contains some fundamental features of the network connectivity ('proper structure') which ensure correct functioning of the system. For example, experimental recordings show that apparently-pathological activity (synchrony) can sometimes appear soon after swimming initiation: the two body sides spike synchronously during several cycles before then switching to normal anti-phase swimming activity (*Li et al., 2014*). This synchronous activity appears also in model simulations with connectivity generated by both the anatomical and the probabilistic models. However, the number of synchronously firing neurons is significantly reduced in probabilistic connectomes.

A second type of apparently pathological activity is the additional firing of some dINs near the middle of the swimming cycle (mid-cycle dINs) (*Li et al., 2014*). Mid-cycle dINs appear in model simulations with both anatomical and probabilistic connectivity. However, the number of such mid-cycle dINs is significantly reduced in probabilistic connectomes: 0.8 and 6.3 for probabilistic and anatomical connectomes, respectively (average according to swimming cycles and 100 simulations).. These results suggest that synchrony and mid-cycle dINs arise from connectivity imperfections and that the generalised connectivity encapsulated in the probabilistic model improves on the imperfection of some individual realisations.

To design the probabilistic model, we use a minimalistic approach. We use the assumption that directed connections are represented by the matrix of independent Bernoulli random variables. One of the strengths of this approach is that it allowed us to analytically calculate some of the graph's characteristics (the mean and standard deviation of in- and out-degrees, heterogeneity coefficients) directly from the probability matrix, without considerations of a particular (generated) connectome. In the case of the anatomical model, we can only compute graph characteristics for a connectome realization. Here, we study how these characteristics relate to particular functional properties of the network. For instance, the average in- and out-degrees were used to predict the swimming period and to find the positions of reliably firing cINs.

The assumption that the Bernoulli variables are independent is a significant limitation of the probabilistic model. One way to overcome this limitation might be the use of more sophisticated probabilistic processes where the random variables corresponding to different connections become dependent (e.g. random Markov field approach).

Computational modelling of the tadpole spinal cord reveals the fundamental features of neuronal connectivity that are responsible for robust swimming generation. Unlike simpler organisms such as *C. elegans*, tadpoles have the potential for significant variation between individuals in terms of neuronal connectivity, as a result of the large number of random processes involved in their development. Despite this variation, the behaviour of individuals is approximately the same, suggesting some fundamental organisational principles common across the species. We adopt the philosophy that, for tadpoles at least, there is a theoretical 'perfect' version of the nervous system with individual random variations from this ideal. Although, the probabilistic model arises from 'averaging' of many anatomical connectomes, this model still generates connectomes that reliably swim and this fact presumably reflects the fundamental organisational principles of the system. An interesting

property of connectomes generated by the probabilistic model is that their anatomical and functional characteristics are considerably less variable than those generated by our anatomical model (and on whose properties the probabilistic model was based). We hypothesise that due to the 'averaging' process of the probabilistic model, the connectomes generated from it are closer to the theoretical 'ideal' network. Some characteristic features of the connectivity are not clear from an individual realisation, but become evident from the probabilistic model. For example, the shape of degree distributions as a function of cell position cannot be clearly seen from analysing an individual connectome – these shapes are irregular. They are much clearer when calculated directly from the probabilistic model itself. In addition to this, connectomes generated by the probabilistic model generate spiking activity that is considerably less variable and 'messy' than anatomical connectomes, which makes it easier to see and quantify phenomena such as irregularly spiking cINs.

Finding neural connection probabilities under biological constraints is a difficult problem. In the case of the tadpole spinal cord, the system is simple enough that it is possible to reconstruct biologically realistic connectivity (*Roberts et al., 2014*) (an anatomical connectome) and to define neuronal connection probabilities (probabilistic model). We believe that this is a promising general approach that could be used beyond the particular case of tadpoles. Similar probabilistic approaches have been used for modelling the development of neural networks using limited experimental data (*Binzegger et al., 2004*; *Zubler and Douglas, 2009*). Another possible approach for finding connection probabilities is to minimize an appropriate cost function which reflects both anatomical and functional properties. A combination of these approaches has been used in pilot studies that aim to incorporate into the tadpole connectome a new group of neurons recently found in the hindbrain (*Buhl et al., 2015*). We will report this result in a separate publication.

## Conclusion

We study the structure and function of the spinal cord neuronal network using experimental data and computational modelling. Our anatomical model generates multiple highly variable and nonhomogeneous connectomes and to deal with this large and complex data we design a very simple mathematical meta-model expecting that this new probabilistic model will reflect (generalise) structural properties of anatomical connectomes and show proper functioning.

The crucial question is: 'Can probabilistic connectomes produce swimming'? The answer to this is not obvious. Our earlier paper (*Li et al., 2007a*) showed that a graph of connections based on probabilities derived from small number of pairwise recordings provides swimming in about 60% of cases only. On the other hand, this new study shows that probabilistic connectomes that include some of the structure of anatomical connectomes reliably swim in all cases. Thus, we can derive an important conclusion that the two properties of the probabilistic model inherited from anatomical connectomes: (1) position of neurons along the rostro-caudal coordinates and (2) frequency of connection appearance, are sufficient for swimming generation.

Also, it is easy to use the probabilistic approach to generate connectomes compared to the need to 'grow' them using the anatomical model: all traditional characteristics of the connectivity graph can be calculated directly from the probability matrix without consideration of particular connectomes. Some characteristics of the probabilistic connectomes (e.g. the mean of in- and out-degrees) coincide with equivalent characteristics of the anatomical connectomes but some differ (e.g. the variances of in- and out-degrees are significantly smaller for probabilistic connectomes). Although there are some differences between the behaviour of anatomical and probabilistic connectomes, even studying these differences can provide important insights into the relationship between the structure and function of the network. Our investigation in the reasons underlying a difference in swimming frequency between the two types of connectome (see result section) is an example of this, where we found that it was the degree of variance of cIN in-degree from dINs that largely caused the difference. It would have been hard to observe this interesting phenomenon without having the probabilistic model (where in-degree variance is much lower) to compare with the anatomical one.

The probabilistic model provides a different way to look at the information generated by the anatomical model. It is grounded in the previous anatomical model as the anatomical model is grounded in the biological anatomy. It provides a different perspective on data generated by many anatomical models, and it is this different perspective that makes the probabilistic model an advance.

## Acknowledgements

We thank Prof Alan Roberts for his valuable contribution to model development, discussion of results and many other useful comments. We also thank Dr Wen-Chang Li for his valuable assistance. This work was supported by the UK Biotechnology and Biological Sciences Research Council (BB/L002353/1; BB/L000814/1). AF was supported by a PhD studentship from Plymouth University. We would like to thank the Reviewing Editor Prof Ronald Calabrese and anonymous reviewers for their fruitful comments. RB is on leave from the Institute of Mathematical Problems of Biology, The Branch of Keldysh Institute of Applied Mathematics of Russian Academy of Sciences.

## Additional information

### Funding

| Funder | Grant reference number | Author |
| --- | --- | --- |
| Biotechnology and Biological Sciences Research Council | BB/L000814/1, BB/L002353/1 | Andrea Ferrario Stephen R Soffe Roman Borisyuk |
| Plymouth University | | Andrea Ferrario |

The funders had no role in study design, data collection and interpretation, or the decision to submit the work for publication.

### Author contributions

Andrea Ferrario, Conceptualization, Data curation, Software, Formal analysis, Funding acquisition, Validation, Investigation, Visualization, Methodology, Writing—original draft, Writing—review and editing; Robert Merrison-Hort, Conceptualization, Supervision, Methodology, Writing—review and editing; Stephen R Soffe, Conceptualization, Resources, Supervision, Funding acquisition, Investigation, Writing—review and editing; Roman Borisyuk, Conceptualization, Resources, Supervision, Funding acquisition, Methodology, Project administration, Writing—review and editing

### Author ORCIDs

Andrea Ferrario http://orcid.org/0000-0001-9082-1555
Robert Merrison-Hort http://orcid.org/0000-0001-8215-7527
Roman Borisyuk https://orcid.org/0000-0003-1384-9057

### Decision letter and Author response

Decision letter https://doi.org/10.7554/eLife.33281.018
Author response https://doi.org/10.7554/eLife.33281.019

## Additional files

### Supplementary files

• Transparent reporting form
DOI: https://doi.org/10.7554/eLife.33281.011

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

# Appendix 1

DOI: https://doi.org/10.7554/eLife.33281.012

## Anatomical model

The anatomical model generates the complete neural connectivity in the spinal cord using a 'developmental' approach that mimics axon growth (**Borisyuk et al., 2014**). The axon growth model is based on numerous anatomical data of *Xenopus* tadpole spinal cord neurons. Our attempt is to include in the model as much biological information as possible. Here, we give a brief description of the model, but full details can be found in (**Borisyuk et al., 2014**; 5, 4).

### The model spatial structure

Tadpole's spinal cord is approximated as a 2D rectangular plate where neuronal bodies, dendrites and axons are located (*Figure 1* in the main text). The third dimension (i.e. the thickness of the spinal cord 'tube') was ignored as this is very thin (10 µm thick). In the model description, variables $x$ and $y$ correspond to the rostro-caudal distance (RC) from the midbrain-hindbrain border and the dorso-ventral distance (DV) from the ventral mid-line, respectively. Positive (Negative) values of $y$ correspond to positions on the left (right) side of the body. We consider a limited area of the spinal cord, where $(x, y) \in [500 \ \mu m, 2000 \ \mu m] \times [-145 \mu m, 145 \mu m]$.

### Axon growth model

We describe the axon growth using discrete time iteration map (with time step 1 ms). The map is described by three variables $(x_n, y_n, \theta_n)$, where $x_n$ represents RC coordinate, $y_n$ the DV coordinate and $\theta_n$ the growth angle of the axon at each time step $n$ $(n = 0, 1, \ldots, N)$. The map for the growth angle depends on a 'stiffness' term, which is the tendency of the growth angle to grow straight, and by the influence of environmental cues (according to chemical gradients), which deviate the growth cone from a straight path. The chemical gradients functions $G_{RC}(x, y)$ and $G_{DV}(x, y)$ depend on the current position of the axon and that will determine the change of the growth angle at each time step on the RC and DV axis, respectively. Additionally, a uniform random variable $\epsilon_n$ is included to provide an additional degree of freedom at each time step. The map is described by the following equations:

$$x_{n+1} = x_n + \Delta cos \ \theta_n \tag{1a}$$

$$y_{n+1} = y_n + \Delta sin \ \theta_n \tag{1b}$$

$$\theta_{n+1} = \theta_n - G_{RC}(x_n, y_n) sin \ \theta_n + G_{DV}(x_n, \ y_n) cos \theta_n + \in_n \tag{1c}$$

Here, the elongation parameter $\Delta = 1 \mu m$; $\epsilon_n$ uniformly distributed in the interval $[-\alpha, \alpha]$. To start the axon growth simulation we assume that the axon initial positions $(x_0, y_0)$ coincide with the soma positions. The initial value for the growth angle $\theta_0$ and the axon length $L = N \cdot \Delta$ (which will determine the number of iterations of the map) are randomly selected from the distribution of experimentally measured initial angles and axon lengths.

The gradient functions $G_{RC}$ and $G_{DV}$ depend on various parameters that describe the properties of the chemical gradients in the 2D space. Previous studies revealed that the axons of the same neuron type tend to grow in specific regions of the spinal cord, suggesting that such axons could be controlled by the same gradients. Thus, the parameters of the gradient functions in the model were selected to reproduce the statistical properties of the axons of each specific cell type separately. Some of these parameters were selected according to general biological knowledge on the distribution and properties of the chemical gradients. The remaining parameters were estimated using an optimization technique that minimizes a custom cost function that measures the similarity between statistical properties of simulated

and experimentally measured axons. For a detailed description of the gradient functions $G_{RC}$ and $G_{DV}$, the cost function and the optimization technique see (**Borisyuk et al., 2014**.

Anatomical and physiological studies in tadpoles have revealed that there are physical constraints to the axons growth, and these have been added in the anatomical model. In particular, longitudinal barriers delimited by DV coordinates $y = \pm 125$ delimit the area where axons of all neuron types except RBs grow (called marginal zone). Coordinates $y \in (127, 137)$ on the left body side ($y \in (-137, -127)$ for the right body side, respectively) delimit the region where RB axons can grow (called dorsal tract area).

## Branching and commissural axons

The axon of tadpole spinal neurons typically splits into two branches during its growth. We call primary axon the branch that starts from the soma and continues to grow following the growth direction before the branching point. Secondary axons grow from the branching and grow towards the opposite direction of the primary axon (rostro-caudally).

The RC direction of primary axon of each neuron type is known from anatomy of neurons. In the model we thus consider growth of primary axons and secondary axons as two consecutive processes. All primary axons grow first, and after that secondary axon growth starts at a specified branching point. The coordinates of branching points are selected randomly from the distribution of available experimental data for each neuron type.

The axon of commissural neurons (dlcs and cINs) starts to grow on the cell body side and then rapidly navigates on the opposite body side after crossing the floor plate due to the influence of strong DV gradients. After crossing, DV gradients change their sensitivity and become weak, and the axon starts to deviate towards ascending direction. Secondary axons are positioned on the contralateral side and grow in the descending direction. At the beginning, commissural neurons grow in the ventral direction according to the axon growth equations with specially adjusted parameter values. After crossing the boundary of the ventral plate on the opposite side the axon growth is described by the same equations but with another regular set of parameter values (for details see (**Borisyuk et al., 2014**)).

## Synapses and full connectome

Dendrites are assumed to be fixed bars extending dorso-ventrally. Thus, each dendrite is represented by a pair of DV positions, one corresponding to its lower (ventral) bound and on to its upper (dorsal) bound. For each neuron, cell body positions and dendritic bounds are sampled from the distributions of experimentally measured data for each specific cell type.

In model simulations the number of neurons of each type is fixed and it is the same for both sides of the body (see methods section in the main text). In reality, although biological data are limited, total numbers of spinal neurons and the population sizes of individual neuron types do not appear to vary greatly between animals (perhaps $\pm 10$–15% at most) at this early stage of development. Any variation that there is in numbers is small compared to the differences from the previous developmental stage and the following stage, both of which swim in the same way.

For each pair of pre- and post- synaptic neurons, a connection is generated whenever the pre-synaptic axon crosses the dendritic bar of the post-synaptic neuron with some probability. Such probability depends on pre and post synaptic cell types and was estimated from various experimental pairwise recordings (for details see (**Borisyuk et al., 2014**)). During the initial pre-crossing stage we assume that the axon of commissural neurons cannot produce synapses.

Since the model uses several number of random variables, each simulation of the anatomical model generates a different connectome.

# Appendix 2

DOI: https://doi.org/10.7554/eLife.33281.013

## Functional model

The probabilistic and anatomical connectomes provides detailed information on connectivity in the spinal cord – specifically a list of synaptic connections between neurons. We use this information to build a functional model that simulates the spiking activity of spinal cord neurons. This allows us to study one of the fundamental problems of neuroscience: the relationship between connection structure and functionality.

## Overview

To simulate the activity in the generated connectomes we represent each cell as a single compartment conductance based neuron of Hodgkin-Huxley type. The equation governing the membrane potential (V) for neuron i is:

$$C\frac{dV_i}{dt}_i = I_{lk} + I_{Na} + I_{Kf} + I_{Ks} + I_{Ca} + I_{syn} + I_{gj} + I_{ext} \tag{2}$$

The capacitance $C$ of all neurons is $10pF$, which corresponds to a density of $1.0\mu F/cm^2$ for a total surface area of $10^{-5}cm^2$. The terms $I_{lk}$, $I_{Na}$, $I_{Kf}$, $I_{Ks}$ and $I_{Ca}$ represent transmembrane currents mediated by different ions, respectively: non-specific leak, sodium, fast potassium, slow potassium and calcium. The terms $I_{syn}$ and $I_{gj}$ represent the summed inputs from chemical synapses ($I_{syn}$) and gap junctions ($I_{gj}$), while $I_{ext}$ is an externally-injected current. Although the different neuron types in the tadpole spinal cord have different electrophysiology, for simplicity we use the model of a motoneuron from for most model neurons, as this shows characteristics (e.g. repetitive firing in response to injected current) that are broadly shared by all of the neuron types. The exception to this is dINs, which have special properties such as only firing a single spike in response to current injection and the ability to fire post-inhibitory 'rebound' spikes. Model dINs differ from non-dINs in the following ways:

- The parameter values governing the membrane properties are different (*Appendix 2—table 2*).
- Only dINs contain a calcium current. For non-dINs, we set $I_{Ca} = 0$.
- Only dINs make gap junction connections (and only with other dINs).

## Membrane channels

The leak, sodium and potassium channel currents are given by the following equations:

$$I_{Na}(t) = hm^3\bar{g}_{Na}(V_i - E_{Na}) \tag{3}$$

$$I_{Kf}(t) = n_f^4\bar{g}_{Kf}(V_i - E_K) \tag{4}$$

$$I_{Ks}(t) = n_s^2\bar{g}_{Ks}(V_i - E_K) \tag{5}$$

The parameters $E_{lk}$, $E_{Na}$ and $E_K$ give the reversal potential for the leak, sodium and potassium channels respectively, and the parameters $\bar{g}_{lk}$, $\bar{g}_{Na}$, $\bar{g}_{Kf}$ and $\bar{g}_{Ks}$ give their maximum conductances (these parameter values are given in *Appendix 2—table 1*). The gating variables $h$, $m$, $n_f$ and $n_s$ are governed by equations (*Equation 6*; *Equation 7*; *Equation 8*; *Equation 9*), where $X = h, m, n_f, n_s$.

$$\tau_X(V)\frac{dX}{dt} = (X_\infty(V) - X) \tag{6}$$

$$X_\infty(V) = \alpha_X(\nu)(\alpha_X(\nu) + \beta_X(\nu))^{-1} \tag{7}$$

$$\tau_X(V) = (\alpha_X(\nu) + \beta_X(\nu))^{-1} \tag{8}$$

$$\alpha_X(V), \beta_X(V) = \frac{A + BV}{C + \exp\left(\frac{D+V}{E}\right)} \tag{9}$$

The values of the parameters A, B, C, D and E in the functions $\alpha_X(V), \beta_X(V)$ were taken from Sautois et al. (2007) for non-dINs and from (*Roberts et al., 2014*) for dINs, and are shown in *Appendix 2—table 2*.

As in (), model dINs contain a calcium-mediated current which is modelled according to the Goldman-Hodgkin-Katz equation. This current is calculated as:

$$I_{Ca} = h_{Ca}^2 p_{Ca} zFx \frac{S_{in} - S_{out}\exp(-x)}{1 - \exp(-x)} \tag{10}$$

$$x = \frac{zFV_i}{RT} \tag{11}$$

Here, $p_{ca}$ is the permeability of the membrane to calcium ions (analogous to maximum conductance) and $z$ is their ionic valence (+2). $S_{in}$ and $S_{out}$ are the concentration of calcium in and outside of the cell, respectively. F is Faraday's constant, and R is the ideal gas constant, while T is the temperature in Kelvin. Parameters of the calcium current are $p_{ca} = 14.25\ cm^3/ms$, $F = 96485\ C/mol$, $R = 8.314 J/(K\ mol)$, $T = 300\ K$, $[Ca^{2+}]_i = 10^{-7}\ mol/cm^3$, $[Ca^{2+}]_o = 10^{-5}\ mol/cm^3$. Finally, $h_{Ca}$ is the gating variable associated with the calcium current, which is governed by the standard gating equations (*Equation 6*; *Equation 7*; *Equation 8*; *Equation 9*) – although note from *Appendix 2—table 2* that two different sets of parameters are used for this equation based on whether the membrane potential is above or below $-25$ mV.

**Appendix 2—table 1.** Maximal conductance (in nS) and equilibrium potential (in mV) of each ionic channel in the model neurons.

|          | $g_{lk}$ | $e_{lk}$ | $g_{Na}$ | $e_{Na}$ | $g_{Kf}$ | $e_{Kf}$ | $g_{Ks}$ | $e_{Ks}$ |
|----------|------|------|--------|------|------|------|------|------|
| dIN      | 1.4  | −52  | 240.5  | 50   | 12   | −80  | 9.6  | −80  |
| non−dIN  | 2.47 | −61  | 110    | 50   | 8    | −80  | 1    | −80  |

DOI: https://doi.org/10.7554/eLife.33281.014

**Appendix 2—table 2.** Parameters defining the rate functions of the model neurons rounded to the first decimal digit for dINs and non-dIN neuronal types (− sign that the cell type has no contribution of the specific channel variable, units of measures are given in the first row of each parameter; parameter *C* is dimensionless).

| dIN / non−dIN | Rate Function | A (ms $^{-1}$) | B (ms$^{-1}$mV$^{-1}$) | C (−) | D (mV) | E (mV) |
|---------------|---------------|---------|----------|---------|-----------|------------|
|               | $\alpha_r$ | 4/− | 0/− | 1/− | −15.3/− | −13.6/− |
| Ca | $\beta_r(\nu < -25mV)$ | 1.2/− | 0/− | 1/− | 10.6/− | 1/− |
|               | $\beta_r(\nu > -25mV)$ | 1.3/− | 0/− | 1/− | 5.4/− | 12.1/− |
| K−fast | $\alpha_f$ | 5.1/3.1 | 0.1/0 | 5.1/1 | −18.4/−27.5 | −25.4/−9.3 |
|        | $\beta_f$ | 0.5/0.4 | 0/0 | 0/1 | 28.7/9 | 34.6/16.2 |
| K−slow | $\alpha_s$ | 0.5/0.2 | 8.2e − 3/0 | 4.6/1 | −4.2/−3 | −12/−7.7 |
|        | $\beta_s$ | 0.1/0.05 | −1.3e − 3/0 | 1.6/1 | 2.1e5/−14.1 | 3.3e5/6.1 |

*Appendix 2—table 2 continued on next page*

*Appendix 2—table 2 continued*

| dIN / non−dIN | Rate Function | A (ms$^{-1}$) | B (ms$^{-1}$mV$^{-1}$) | C (−) | D (mV) | E (mV) |
|---|---|---|---|---|---|---|
| Na | $\alpha_m$ | 8.7/13.3 | 0/0 | 1/0.5 | −1/−5. | 12.6/−12.6 |
| | $\beta_m$ | 3.8/5.7 | 0/0 | 1/1 | 9/5 | 9.7/9.7 |
| | $\alpha_h$ | 0.1/0.04 | 0/0 | 0/0 | 38.9/28.8 | 26/26 |
| | $\beta_h$ | 4.1/2 | 0/0 | 1/1e−3 | −5.1/−9.1 | −10.2/−10.2 |

DOI: https://doi.org/10.7554/eLife.33281.015

## Synaptic currents

The synaptic current that arises in a neuron is the combination of three different sub-types of synaptic receptor: excitatory AMPA and NMDA and inhibitory glycine:

$$I_{syn} = I_{ampa} + I_{nmda} + I_{inh} \tag{12}$$

Each synaptic current is calculated using the following equation, where $X = ampa, \ nmda, \ inh$:

$$I_x = \sum_j \left\{ \bar{g}_{i,j}^X f_X(V_i) \sum_{s \in S_j(t)} \Delta_X \left( \exp\left( \frac{s + \delta_{i,j} - t}{\tau_c^X} \right) - \exp\left( \frac{s + \delta_{i,j} - t}{\tau_o^X} \right) \right) \right\} \tag{13}$$

Here, $\bar{g}_{i,j}^X$ is the maximum conductance ('strength') of synaptic connection of type X from neuron $i$ to neuron $j$. If the connectome does not include a connection from $i$ to $j$ then $\bar{g}_{i,j}^X = 0$, otherwise it is selected according to the type of the pre- and post- synaptic neurons, based on paired recordings. Pre-synaptic neuronal type determines the synapse type $X$. Inhibitory neuron types are cIN and aIN; excitatory ones are the remaining cell types. The synaptic strengths used in the model are typically the ones given in **Appendix 2—table 3**, except for few values that were modified to match the physiology of neurons and synapses (details are given in (**Roberts et al., 2014**)). Specifically, the maximal conductance of AMPA synapses from RBs to dli neurons are set to $8nS$, the maximal conductance of AMPA synapses from dINs to aINs are set to $0.1nS$, the maximal conductance of NMDA synapses from dIN to dINs are set to $0.15nS$, and the maximal conductance of NMDA synapses from RB to dlc are set to 1nS..

The set $S_j(t)$ contains the times of all the spikes that neuron $j$ has fired up to the current time $t$. Each spike generates a post-synaptic current (PSP) that rises according to the time constant $\tau_o^X$ and decays according to $\tau_c^X$. The normalizing constant $\Delta_X$ is set such that the peak of the sum of the exponentials is 1, meaning that following a spike the conductance rises to a maximum of $\bar{g}_{i,j}^X$. The values selected for the time and normalizing constants are given in **Appendix 2—table 3** and they are based on previous modelling (**Roberts et al., 2014**. To mimic synaptic strength variability, Gaussian noise with standard deviation 5% of the mean was added to the maximum conductance of each individual synapse.

**Appendix 2—table 3.** Parameters of the synaptic models.

| X | NMDA | AMPA | INH |
|---|---|---|---|
| $\tau_o^X (ms)$ | 0.5 | 0.2 | 1.5 |
| $\tau_c^X (ms)$ | 80 | 3.0 | 4.0 |
| $\Delta_X (-)$ | 1.25 | 1.25 | 3.0 |
| $E_X (mV)$ | 0 | 0 | −75 |
| $g_X (nS)$ | 0.29 | 0.593 | 0.435 |

DOI: https://doi.org/10.7554/eLife.33281.016

The synaptic delay between two neurons, $\delta_{i,j}$, consists of a constant and distance-dependent part:

$$\delta_{i,j} = \delta_C + \delta_D |P_i - P_j|$$

Here, $P_i$ and $P_j$ are the positions of neurons $i$ and $j$ along the rostro-caudal axis, $\delta_C$ is the constant delay and $\delta_D$ is the speed of synaptic transmission. We set $\delta_C = 1ms$ and $\delta_D = 0.0035ms/\mu m$.

Finally, the function $f_X(V)$ determines how the synaptic current depends on the post-synaptic voltage. For AMPA and inhibitory synapses this has a simple linear (Ohmic) form:

$$f_X(V) = E_X - V$$

Where $X = ampa, inh$ and $E_X$ is the equilibrium (reversal) potential of the synapse type (**Appendix 2—table 3**). As a result of magnesium block, NMDA synapses have an additional non-linear voltage dependence, which we include by adding a sigmoidal scaling term to $f_{NMDA}$:

$$f_{NMDA}(V) = (E_{nmda} - V)(1 + 0.05 \cdot \exp(-0.08 \cdot V))^{-1}$$

## Gap junctions

Descending interneurons (dINs) are electrically coupled to other nearby dINs via gap junctions. For dINs only, the gap junction current is calculated using a simple Ohmic relationship:

$$I_{gj} = \sum_{j \in G_i} \bar{g}_{gj} (V_j - V_i)$$

Here $G_i$ is the set of indexes of all dINs that are on the same side of the body as neuron $i$ and are located within $D_{gj}$ of neuron $i$ on the rostro-caudal axis, where we set $D_{gj} = 100\mu m$. The parameter $\bar{g}_{gj}$ gives the conductance of gap junctions, and we use the value $\bar{g}_{gj} = 0.2nS$.

