## [Decision Letter]

Thank you for submitting your article "Structural and functional properties of a probabilistic model of neuronal connectivity in a simple locomotor network" for consideration by *eLife*. Your article has been reviewed by three peer reviewers, one of whom, Ronald Calabrese is a member of our Board of Reviewing Editors and the evaluation has been overseen by Timothy Behrens as the Senior Editor.

The reviewers have discussed the reviews with one another and the Reviewing Editor has drafted this decision to help you prepare a revised submission.

Summary:

This manuscript reports on a modeling approach for building network models based on connectomics data. Using a developmental (anatomical) model of the *Xenopus* hatchling tadpole spinal cord connectome, the authors generated 1,000 conectomes from which they derive a complete probability matrix of connections in the spinal cord. From this probability matrix (probabilistic model) they then generate ensembles of 100 connectomes (model instances) to simulate based on previous functional models of swimming activity. They find that these instances of the probabilistic model are indeed functional and more robust that models based on developmental model connectomes, which they attribute to lower variance among connectomes for the probabilistic model. They then analyze these model instances to determine functional properties of the swim network, for example showing that the network does not appear to have hub neurons as an organizing principle and explicating how activity in a hemi-network arises. The technique may applicable to connectomics data and provide valuable insights that could benefit both modelers and experimentalists in the field.

Essential revisions:

The reviewers had partially overlapping concerns, which we have tried to combine as much as possible, but some overlap will still be noted in the required revisions below that represent a strong consensus. All minor concerns from the expert reviewers should also be addressed.

1) Both introduction and discussion mention the probabilistic model extracts fundamental structural features. However, the model consists of a cell-level probability matrix, which suggests these structural features may not be so fundamental but specific to the particular assumptions in this circuit model. Furthermore, since the probabilistic model is derived from another model (the anatomical) and not directly from experimental data, it is not clear whether it might just be capturing properties of this anatomical model that don't really generalize to the real system. Moreover, because the probabilistic description is cell-wise, it may just reflect an overfitting to the anatomical model. The authors should provide further justification of (1) how well the anatomical model captures the real system, and (2) why/how the probability model is capturing fundamental structure features, despite being at a cell-level granularity. This will also help determine how significant/valid the analyses and predictions made by the model are.

2) The analysis of generative models of neuronal networks is a promising avenue to explore links between development, structure and function. However, we find it difficult to understand the results of this paper relative to the Roberts et al., 2014 work on which it builds. We encourage the authors to either restructure the paper in a way that clarifies not only the differences from the previous model, but the advances in the current work and whether they arise from new computational experiments or the new modeling approach.

Once this clarification is achieved then it will be important to show the significance of some of the conclusions. For example, the fact that all connectomes achieve swimming or "this pathological mid-cycle spiking is significantly reduced in probabilistic connectomes" need to be contextualized. Since the probability matrix was directly extracted by averaging over anatomical model instances (all of which showed swimming and no pathologies), perhaps it is not surprising that they show similar properties.

3) While the Bernoulli trial formulation in principle admits more analytical analysis, it is primarily used to compute in- and out-degrees that could just as easily be computed from direct consideration of the results of the anatomical model. As a result, the probabilistic model is a particularly complex null model that captures many aspects of dynamics and not others, but in ways that are only weakly controlled.

In particular, it is not clear why it is better to analyze the ensemble average properties of individual nodes from the probabilistic model rather than analyze the distribution of results from the previous anatomical model. The strongest observation is in subsection “Functional properties of the model: reliable swimming”, effectively relating the swim period to the variance in in-degree of cINs. In the probabilistic model, the use of a binomial distribution implicitly links mean and variance in a way that clearly isn't true of the anatomical model and is shown meaningful impact on the resulting dynamics. This result could be greatly strengthened, if the authors used a random process where variance could be defined separately from the mean and thus this relationship explored explicitly.

4) The last paragraph in the Discussion section makes it seem as if the authors are proposing a novel methodology. However, this novelty is not clear since employing probability matrices to define models is a standard practice: it’s a feature included in several major neural simulators (NEST, Brian, Moose,.…), and many existing models (e.g. on ModelDB) employ prob matrices, either at a population-level or cell-level (for small networks and when enough exp data is available).

5) Authors should include more details about the functional model employed since this is crucial for understanding the results (e.g. low connection weights can drastically alter the effect of high connection probabilities). A table with the main parameters used in the functional model should be included. Additionally, authors must share all the model and analysis code -- this is common practice nowadays in computational neuroscience to ensure replicability and reproducibility.

[Editors' note: further revisions were requested prior to acceptance, as described below.]

Thank you for resubmitting your work entitled "Structural and functional properties of a probabilistic model of neuronal connectivity in a simple locomotor network" for further consideration at *eLife*. Your revised article has been favorably evaluated by Timothy Behrens (Senior editor), a Reviewing editor, and two reviewers.

The manuscript has been improved but there are some remaining issues that need to be addressed before acceptance, as outlined below:

1) The explanation of why the probabilistic model is an advance over the anatomical model is not yet convincing. Looking at both the response to the reviewers and the argument in the Introduction, it still seems that the same results could have been obtained by consideration of the distribution of networks generated by the anatomical model. The authors must clarify why they emphasize the need to construct the probabilistic matrix as a new step in the analysis process, otherwise averaging the outcome of an existing model and putting into a simple probabilistic framework is a modest and intuitive extension of the previous work without added value.

2) The authors have not fully addressed the issue of why the cell-level probabilistic model is capturing general/fundamental structural features of the tadpole network and not overfitting.

Without addressing these two issues more thoroughly, the impact of the paper rides on its scientific results, connecting the structure of the model to key aspects of its functional output and the authors' emphasis on method is misplaced. Some clarifying points from the expert reviews are presented below.

Reviewer #3:

I don't think the authors have addressed the issue of why the cell-level probabilistic model is capturing general/fundamental structural features of the tadpole network and not overfitting. They argue that it is the result of averaging over 1000 anatomical model connectomes, but these are still a model, e.g. they have a fixed number of neurons for each population, which is not the case for real tadpole networks, and so describing the structure at the cell-level prevents it from being applicable to specimens with a different number of neurons. They also argue that all connectomes achieve reliable swimming, but I fail to see how this is relevant. I think the paper still has value despite this limitation, but I think it should be clearly stated in the paper.

---

## [Author Response]

Essential revisions:The reviewers had partially overlapping concerns, which we have tried to combine as much as possible, but some overlap will still be noted in the required revisions below that represent a strong consensus. All minor concerns from the expert reviewers should also be addressed.1) Both introduction and discussion mention the probabilistic model extracts fundamental structural features. However, the model consists of a cell-level probability matrix, which suggests these structural features may not be so fundamental but specific to the particular assumptions in this circuit model. Furthermore, since the probabilistic model is derived from another model (the anatomical) and not directly from experimental data, it is not clear whether it might just be capturing properties of this anatomical model that don't really generalize to the real system. Moreover, because the probabilistic description is cell-wise, it may just reflect an overfitting to the anatomical model. The authors should provide further justification of (1) how well the anatomical model captures the real system, and (2) why/how the probability model is capturing fundamental structure features, despite being at a cell-level granularity. This will also help determine how significant/valid the analyses and predictions made by the model are.

We agree that this comment is very crucial for understanding the key message of the manuscript and we would like to thank the reviewers for important and fruitful suggestions. Our response and summary of revision related to these comments are explained below. We provide additional description on how the anatomical model captures the real system in the Appendix.

1) The anatomical model captures several important structural properties of the real biological object.

Firstly, in the Conclusion section we have shown that the anatomical model is biologically realistic. The model mimics the growth of axons inside a region of the spinal cord, and parameters of the model are optimized to produce the same statistical characteristics as the experimentally measured axons. Moreover, the model uses numerous biological data (including location of neuron bodies, dendritic bounds, and axon properties) and general anatomical information known from experiments.

Secondly, in the Discussion section we have shown that the connection architecture generated by the anatomical model can be combined with a functional model of spiking units to demonstrate very reliable swimming activity, similar to the one observed experimentally.

Relevant changes in the Introduction (also look at Appendix):

“This anatomical model mimics the realistic growth of axons in the spinal cord. Following biological realism, the axon growth is guided by the concentration of chemical gradients in the spinal cord. The properties of such gradients are controlled by model parameters that have been optimized to produce the same statistical characteristics as real measurements. Other model specifications (including soma positions and dendritic extents) are assigned from the distributions of experimental data and from general biological knowledge”

2) The probabilistic model (matrix) for each pair of neurons gives the probability of directed connection. Each probability is the result of averaging 1000 connectomes generated by the anatomical model. Thus, despite having cell-level granularity, the probabilistic model reflects the biological structure of the real system, because it is generated from the anatomical connectomes.

The probabilistic model generates connectomes and the fundamental feature of these probabilistic connectomes is that the functional model with any probabilistic connectome demonstrates very reliable swimming. Moreover, comparison of swimming firing patterns produced by anatomical and probabilistic connectomes shows that in the case of the probabilistic connectome the firing pattern is closer to the “ideal” swimming pattern. For example, there is a reduction in pathological activities, such as less mid-cycle dIN firings.

Relevant changes in the Introduction:

“Being derived from multiple biologically realistic (anatomical) connectomes, the probabilistic model reflects the anatomical structure of the biological system.”

“Multiple functional simulations of probabilistic connectomes demonstrated a reliable pattern of rhythmic activity, qualitatively like tadpole swimming and as seen in previous modelling (Roberts.et al., 2014). Thus, the generalised probabilistic model shares structural and functional properties with the real biological object.”

Relevant changes in the Discussion section:

“This synchronous activity appears also in model simulations with connectivity generated by both the anatomical and the probabilistic models. […].These results suggest that synchrony and mid-cycle dINs arise from connectivity imperfections and that the generalised connectivity encapsulated in the probabilistic model improves on the imperfection of some individual realisations.”

3) We find that in the probabilistic model the variability of in- and out- degrees is less than in the anatomical model and analyse the consequences of this (comparison of swimming periods in case of probabilistic and anatomical connectomes).

Relevant changes the Introduction:

“Specifically, we found the number of incoming connections (in-degree) or out-going connections (out-degree) of each neuron is higher in anatomical rather than probabilistic connectomes. As a result of this finding, we observed that the period of the rhythm was longer in probabilistic connectomes.”

We appreciate the reviewers’ concern about overfitting. However, the probabilistic model estimates probability of each connection separately and independently from other connection probabilities. Each estimate is based on 1000 cases/observations from the anatomical model.

Relevant changes the Introduction:

“It is important to note that the anatomical model provides a way of generating many different connectomes, such that the random variation observed between generated connectomes has the same statistical properties as measurements taken from different individual animals. This helps to avoid the problem of overfitting the data, as we were able to show that all generated connectomes were able to produce reliable swimming.”

2) The analysis of generative models of neuronal networks is a promising avenue to explore links between development, structure and function. However, we find it difficult to understand the results of this paper relative to the Roberts et al., 2014 work on which it builds. We encourage the authors to either restructure the paper in a way that clarifies not only the differences from the previous model, but the advances in the current work and whether they arise from new computational experiments or the new modeling approach.Once this clarification is achieved then it will be important to show the significance of some of the conclusions. For example, the fact that all connectomes achieve swimming or "this pathological mid-cycle spiking is significantly reduced in probabilistic connectomes" need to be contextualized. Since the probability matrix was directly extracted by averaging over anatomical model instances (all of which showed swimming and no pathologies), perhaps it is not surprising that they show similar properties.

This paper describes a new probabilistic model, which is based on our previous work. All our results follow from new modelling and numerous computational experiments. We revised and added the text below to clarify this statement and highlight the significance of our conclusions.

1) In Roberts et al., 2014 we reported both the anatomical model (to generate connectomes) and the functional model (to produce spiking activity of the spinal cord). In the current manuscript, we describe a new probabilistic model, which also generates connectomes. First, we show that the probabilistic model generates connectomes with properties that match the biological object. Second, we demonstrate several important advantages of the probabilistic model in comparison with the anatomical one. Third, we show that the probabilistic model allows us to explain some experimental data. For example, the probabilistic model reveals how the position of cIN neurons is correlated to their firing reliability.

Relevant changes in the introduction:

“Despite the differences between anatomical and probabilistic models, we demonstrate several important advantages of using the probabilistic model in comparison to the anatomical one. For example, we could predict the position of commissural interneurons (cINs) that are active during swimming, which cannot be explained by the anatomical model. Specifically, our simulations show that cINs in rostral positions are less likely to fire reliably than those in caudal positions. Moreover, the probabilistic model allowed us to easily design new computational experiments that helped to clarify the following experimental findings.”

2) Although both the probabilistic and anatomical connectomes generate reliable swimming, the anatomical connectome in many cases produces mid-cycle dIN spiking and synchrony at the start of a swimming episode which are pathological activities.Therefore, in this respect the probabilistic model is superior to the anatomical one and generates an activity pattern which is closer to the ideal swimming pattern. To better demonstrate this conclusion and provide additional context we have added some results on comparison of motor neuron spike reliability.

Relevant changes in the Discussion section:

“This synchronous activity appears also in model simulations with connectivity generated by both the anatomical and the probabilistic models. […]. These results suggest that synchrony and mid-cycle dINs arise from connectivity imperfections and that the generalised connectivity encapsulated in the probabilistic model improves on the imperfection of some individual realisations,”

3) While the Bernoulli trial formulation in principle admits more analytical analysis, it is primarily used to compute in- and out-degrees that could just as easily be computed from direct consideration of the results of the anatomical model. As a result, the probabilistic model is a particularly complex null model that captures many aspects of dynamics and not others, but in ways that are only weakly controlled.In particular, it is not clear why it is better to analyze the ensemble average properties of individual nodes from the probabilistic model rather than analyze the distribution of results from the previous anatomical model. The strongest observation is in section 3.2, effectively relating the swim period to the variance in in-degree of cINs. In the probabilistic model, the use of a binomial distribution implicitly links mean and variance in a way that clearly isn't true of the anatomical model and is shown meaningful impact on the resulting dynamics. This result could be greatly strengthened, if the authors used a random process where variance could be defined separately from the mean and thus this relationship explored explicitly.

In the current manuscript, designing the probabilistic model, we use the minimalistic approach to construct as simple model as possible. Therefore, we use the assumption that directed connections are represented by a matrix of independent Bernoulli variables. This theoretical assumption allows us analytically to calculate some of the graph’s characteristics (the mean and the variance of in- and out-degrees, heterogeneity coefficient) without consideration of generated connectomes. For example, the distribution of in-degrees is described by the Poisson binomial distribution and it allows us to use formulas for the mathematical expectation and the variance of in-degrees. We have modified Figure. 3A to show the standard deviation of in-degree of the probabilistic model.

In case of anatomical model, we can only compute the graph’s characteristics for a particular realization of connections.

We agree that the result can be strengthened in case of using the more sophisticated random model. However, here we show that our approach with the simplest probabilistic model provides fruitful results.

Below are reported the parts added in the manuscript to answer to this point.

Relevant changes in the Discussion section:

“To design the probabilistic model, we use a minimalistic approach. We use the assumption that directed connections are represented by the matrix of independent Bernoulli random variables. […] One way to overcome this limitation might be the use of more sophisticated probabilistic processes where the random variables corresponding to different connections become dependent (e.g. random Markov field approach).”

4) The last paragraph in the Discussion makes it seem as if the authors are proposing a novel methodology. However, this novelty is not clear since employing probability matrices to define models is a standard practice: its a feature included in several major neural simulators (NEST, Brian, Moose,.…), and many existing models (e.g. on ModelDB) employ prob matrices, either at a population-level or cell-level (for small networks and when enough exp data is available).

We agree that the last paragraph was not clearly formulated and have adjusted it. We also agree that specification of connection probabilities is a simple and general idea which was used by several authors and neural simulators for generating neuronal connectivity. The novelty of probabilistic model is that it can be used to compute connection probabilities. In fact, we have three statements here:

1) Tadpole’s spinal cord is a unique case for the study of neural connectivity, which allows us to develop both a biologically realistic connectome and its generalization, the probabilistic model.

2) Generally, finding the biologically realistic matrix of connection probabilities is a very difficult problem. There are several publications attempting to define the probabilities of connection using limited biological data and we provide references to these publications at the last paragraph of Discussion section.

3) We discuss how to use the nonlinear optimization technique to find the connection probabilities under some biological constraints. To adjust probabilities, a cost function can be derived to take into account biological limitations. Optimization of this cost function will provide the best probability values.

We replaced the last paragraph of the Discussion section with the following:

“Finding neural connection probabilities under biological constraints is a difficult problem. […] We will report this result in a separate publication.”

5) Authors should include more details about the functional model employed since this is crucial for understanding the results (e.g. low connection weights can drastically alter the effect of high connection probabilities). A table with the main parameters used in the functional model should be included. Additionally, authors must share all the model and analysis code -- this is common practice nowadays in computational neuroscience to ensure replicability and reproducibility.

Details about the anatomical and functional model formulation and parameters were added in the Appendix.

We provided the code for running all the simulations of the anatomical, probabilistic and functional models in ModelDB. We have temporarily made the code private, but we will make it public once the work has been published. Even though the code is private, it can be downloaded using a read-only “access code”. To download the code, use the link and access code given below:

https://senselab.med.yale.edu/ModelDB/enterCode.cshtml?model=238332

Access Code: tadpole1

The folder “spinal-cord-master/functional model/paper figures” contains the files for generating the figures in the paper.

To run these files, it is required to first generate a bunch of anatomical connectomes, the probabilistic matrix and the functional model. To do so:

1) Enter in the tadpole-spinal-cord-master/anatomical connectome folder and run the MATLAB file run_many_connectomes.m specifying the number of anatomical connectomes to generate (this will generate output anatomical connectome files is the folder “connectome files”).

2) Run the Python file tadpole-spinal-cord-master/probabilistic connectome/build_probabilistic_model.py, specifying the number of anatomical connectomes to use for the probabilistic model. This will automatically generate the probabilistic and anatomical model files in spinal-cord-master/probabilistic connectome/anatomical adjacency matrixes.

3) Enter the folder “spinal-cord-master/functional model/” and compile the nrnmod files in the nrnMod folder (via nrnivmodl Neuron command). The file main.py is the main for launching the simulations and generate output files.

For each of the models a README file contains additional explanations on how to run the simulations

[Editors' note: further revisions were requested prior to acceptance, as described below.]

The manuscript has been improved but there are some remaining issues that need to be addressed before acceptance, as outlined below:1) The explanation of why the probabilistic model is an advance over the anatomical model is not yet convincing. Looking at both the response to the reviewers and the argument in the Introduction, it still seems that the same results could have been obtained by consideration of the distribution of networks generated by the anatomical model. The authors must clarify why they emphasize the need to construct the probabilistic matrix as a new step in the analysis process, otherwise averaging the outcome of an existing model and putting into a simple probabilistic framework is a modest and intuitive extension of the previous work without added value.

We are grateful to all reviewers for these fruitful comments, which clearly show that the motivation and advance of the probabilistic model should be better explained:

The probabilistic model provides a different way to look at the information generated by the anatomical model. It is grounded in the previous anatomical model as the anatomical model is grounded in the biological anatomy. It provides a different perspective on data generated by many anatomical models. Connectivity analysis could be performed directly on results of the anatomical model, however, the anatomical realisations are highly detailed and variable making it actually very difficult to analyse their structure and determine what is important and what is not. In the probabilistic meta-model all connections are independent, again aiding analysis of the generalised structure and allowing identification of core properties. The probabilistic model is based on multiple anatomical networks and simplified by picking up only two elements (the order of neurons along the rostro-caudal dimension and the contact probabilities) to generalise their structural properties. It is the different perspective that makes the probabilistic model an advance.

Also, we believe that the following text highlights the main advances of the probabilistic model and we added this text after the Discussion section as a new “Conclusion” section:

“We study the structure and function of the spinal cord neuronal network using experimental data and computational modelling. Our anatomical model generates multiple highly variable and nonhomogeneous connectomes and to deal with this large and complex data we design a very simple mathematical meta-model expecting that this new probabilistic model will reflect (generalise) structural properties of anatomical connectomes and show proper functioning. […]

et al.,The probabilistic model provides a different way to look at the information generated by the anatomical model. It is grounded in the previous anatomical model as the anatomical model is grounded in the biological anatomy. It provides a different perspective on data generated by many anatomical models, and it is this different perspective that makes the probabilistic model an advance.”

2) The authors have not fully addressed the issue of why the cell-level probabilistic model is capturing general/fundamental structural features of the tadpole network and not overfitting.

See our responses to reviewer #3 comment

Without addressing these two issues more thoroughly, the impact of the paper rides on its scientific results, connecting the structure of the model to key aspects of its functional output and the authors' emphasis on method is misplaced. Some clarifying points from the expert reviews are presented below.Reviewer #3:I don't think the authors have addressed the issue of why the cell-level probabilistic model is capturing general/fundamental structural features of the tadpole network and not overfitting. They argue that it is the result of averaging over 1000 anatomical model connectomes, but these are still a model, e.g. they have a fixed number of neurons for each population, which is not the case for real tadpole networks, and so describing the structure at the cell-level prevents it from being applicable to specimens with a different number of neurons. They also argue that all connectomes achieve reliable swimming, but I fail to see how this is relevant. I think the paper still has value despite this limitation, but I think it should be clearly stated in the paper.

1) See our response to reviewing editor comment 1.

2) Fixed number of neurons:

Currently, in the anatomical model, neuron population sizes and their broad rostro-caudal distributions are fixed, based as far as possible on biological estimates, and are symmetrical between the two sides. Everything else varies from one connectome generation to another, again based on biological measurements: the exact rostro-caudal position of each soma, and therefore the start position of each axon; axon outgrowth angle, the exact trajectory followed, and the length of each axon. We have assumed that the main variation in the distribution of connections will result from these differences in the patterns of axon growth rather than differences in soma numbers, which we expect to be relatively similar between individual animals. Our “averaging” of connectomes should therefore have covered the main variability that would be expected between individuals.

In reality, although biological data are limited, total numbers of spinal neurons and the population sizes of individual neuron types do not appear to vary greatly between animals (perhaps ± 10-15% at most) at this early stage of development.

Any variation that there is in numbers is small compared to the differences from the previous developmental stage and the following stage, both of which swim in the same way.

To address this comment, we included a short clarification to Appendix 1 (subsection “Branching and commissural axons”) related to the anatomical model.

3) Overfitting:

We confess to being a little unsure about the use of the term “overfitting” here. According to our understanding, overfitting occurs when the number of parameters is large, and the amount of data is not sufficient for parameter estimation, meaning that one set of possible parameters is chosen when in fact many would produce the same fit to the data. In the probabilistic model, the number of parameters is of order 1500*1500 (number of connection probabilities), whereas the amount of data for estimation is about 1500*1500*1000 (number of pairs of neurons across 1,000 anatomical connectomes). The probabilistic model is therefore, according to our understanding, a proper generalization of the data that does not suffer from overfitting. If the reviewer can provide a clearer description of their concern, we will be happy to address it in more detail. Our attempt to address the overfitting matter was clearly not successful and has led to confusion, therefore we deleted the sentence from the Introduction.

4) Probabilistic model is capturing general /fundamental structural features:

We agree that our formulation is confusing. In fact, the fundamental feature is the ability to generate reliable swimming by all probabilistic connectomes. The text (Introduction) has been adjusted.